# An original potentiating mechanism revealed by the cryo-EM structures of the human α7 nicotinic receptor in complex with nanobodies

Marie S. Prevost [1,9] ✉, Nathalie Barilone[1,9], Gabrielle Dejean de la Bâtie[1], Stéphanie Pons[2], Gabriel Ayme [3], Patrick England [4], Marc Gielen[1,5], François Bontems[6,7], Gérard Pehau-Arnaudet [8], Uwe Maskos[2], Pierre Lafaye[3] & Pierre-Jean Corringer[1] ✉

The human α7 nicotinic receptor is a pentameric channel mediating cellular and neuronal communication. It has attracted considerable interest in designing ligands for the treatment of neurological and psychiatric disorders. To develop a novel class of α7 ligands, we recently generated two nanobodies named E3 and C4, acting as positive allosteric modulator and silent allosteric ligand, respectively. Here, we solved the cryo-electron microscopy structures of the nanobody-receptor complexes. E3 and C4 bind to a common epitope involving two subunits at the apex of the receptor. They form by themselves a symmetric pentameric assembly that extends the extracellular domain. Unlike C4, the binding of E3 drives an agonist-bound conformation of the extracellular domain in the absence of an orthosteric agonist, and mutational analysis shows a key contribution of an N-linked sugar moiety in mediating E3 potentiation. The nanobody E3, by remotely controlling the global allosteric conformation of the receptor, implements an original mechanism of regulation that opens new avenues for drug design.

Nicotinic acetylcholine receptors (nAChRs) mediate cellular communication in neuronal and non-neuronal cells. In humans, sixteen genes encode for nAChRs subunits that can assemble into homo- or heteropentamers, generating a wide diversity of combinations, each displaying a unique expression pattern and physiological function[1]. Among nAChRs, the homopentameric α7 has attracted considerable interest with the aim of developing agonists or potentiators for the treatment of neurological and psychiatric disorders associated with cognitive decline, such as Alzheimer's disease, schizophrenia, autism or bipolar troubles[2]. The α7-nAChR is also an essential component of the cholinergic anti-inflammatory pathway[3].

nAChRs are pentameric ligand-gated ion channels (pLGIC). Acetylcholine (ACh) binding to the orthosteric site within their extracellular domain (ECD) promotes the fast transition from a resting closed-channel state to an active open-channel state selective for cations. This is followed by a slower transition to a desensitized closed-channel

[1]Institut Pasteur, Université Paris Cité, CNRS UMR 3571, Channel-Receptors Unit, Paris, France. [2]Institut Pasteur, Université Paris Cité, CNRS UMR 3571, Integrative Neurobiology of Cholinergic Systems Unit, Paris, France. [3]Institut Pasteur, Université Paris Cité, CNRS UMR 3528, Antibody Engineering Platform, Paris, France. [4]Institut Pasteur, Université Paris Cité, CNRS UMR 3528, Molecular Biophysics Platform, Paris, France. [5]Sorbonne Université, Paris, France. [6]Institut Pasteur, Université Paris Cité, CNRS UMR 3569, Structural Virology Unit, Paris, France. [7]Institut de Chimie des Substances Naturelles, Centre National de la Recherche Scientifique, Université Paris Saclay, Gif-sur-Yvette, France. [8]Institut Pasteur, Université Paris Cité, Ultrastructural Bioimaging Core Facility, Paris, France. [9]These authors contributed equally: Marie S. Prevost, Nathalie Barilone. ✉e-mail: marie.prevost@pasteur.fr; pierre-jean.corringer@pasteur.fr

state. α7-nAChR specific agonists and partial agonists have been developed, some of which display pro-cognitive effects in pre-clinical trials[4]. Positive allosteric modulators (PAMs) binding in the transmembrane domain (TMD) also showed beneficial effects, as exemplified by the Type II PAM PNU-120596 (PNU) which potentiates the ACh-elicited currents and strongly impairs desensitization[5].

Several human α7-nAChR high-resolution structures have recently been solved by cryo-electron microscopy (cryoEM), in detergent micelles or in protein/lipid nanodiscs[6,7]. The α7-nAChR harbors a β-folded extracellular domain (ECD) carrying the orthosteric site at the interface between subunits, coupled to a TMD with four α-helices per subunit (termed M1 to M4) that form the ion channel, and an intracellular domain (ICD), which contains two helices (MA and MX) flanking a highly variable region which is truncated in the constructs used for solving cryo-EM structures. Closed-channel structures in Apo and antagonist (bungarotoxin (Bgt)-bound conditions have been annotated as resting conformations, while those in complex with agonists (epibatidine or EVP-6124 abbreviated as EVP) have been annotated as desensitized conformations. Structures in complex with both agonist and PNU show an apparently open (Epi) or semi-open (EVP) pore, consistent with an active and partially active state, respectively[8].

The comparison of the resting and open or semi-open conformations shows that agonist binding promotes a global conformational change in the ECD, involving notably a contraction of the orthosteric site with a large inward movement of the C-loop that caps the binding cavity. This is associated with a twist and tilt of each subunit β-sandwich. The tilt moves the upper part of the subunit ECD toward its neighboring complementary subunit while its lower part moves away from it. In consequence, the cys-loop at the bottom of the ECD undergoes an outward movement associated with TMD expansion, M2 tilt and channel opening. From this open-channel conformation, desensitization motions mainly concern the TMD that relaxes back toward an occluded, closed-channel conformation associated with a subtle global expansion of the ECD.

In a previous study, we immunized alpacas with cells expressing an α7/5HT$_3$ chimera to generate highly potent and α7-specific modulators[9]. Among the resulting camelid antibody fragments called nanobodies, the one called E3 acts as a PAM. In two-electrode voltage-clamp electrophysiology (TEVC), upon pre-application for 30–60 s, E3 potentiates ACh-gated currents with a maximal effect around an EC$_{10}$ concentration of ACh of 30 μM. E3 does not alter significantly the EC$_{50}$ of ACh, nor the apparent slow desensitization component of the ACh-elicited response. In contrast, the nanobody called C4 acts as a silent allosteric ligand (SAL), showing no effect on ACh-gated currents but inhibiting E3-elicited potentiation. Both nanobodies do not displace Bgt in binding experiments, pointing to an allosteric binding site away from the orthosteric site.

In this paper, we aim to understand the mechanism by which the nanobodies mediate their allosteric effect by solving the cryo-EM structures of the complexes they form with the α7-nAChR. Our data enables the high-resolution identification of the common binding epitope targeted by C4 and E3, with, in both cases, an association of five nanobodies binding at the top of the receptor pentamer. Structures show a marked action of E3 on the ECD of α7 in the absence of an agonist, and mutational investigations hint at a key N-linked sugar moiety in mediating E3 potentiation.

## Results

### Production of the purified human α7-nAChR reconstituted in nanodiscs

To express the α7-nAChR in mammalian cells, we used a lentivirus-based strategy[10] (Supplementary Fig. 1). We designed two lentiviral constructs harboring a gene coding for either the full-length α7-nAChR subunit fused to a Rho1D4 tag at its C-terminus (α7FLcryo) or for the same gene where the coding sequence of the ICD is replaced by a short

linker based on an α4β2 construct used for crystallogenesis[11] (α7ΔICDcryo) (Supplementary Fig. 2). We additionally produced a lentiviral construct harboring the coding sequence of the chaperone NACHO[12]. Lentiviruses particles were used to infect T-REx cells grown in suspension. Expressed proteins were affinity purified in dodecyl-maltoside (DDM). As initial micrographs in detergent micelles showed protein aggregation, we further reconstituted the protein in MSP1E3D1 nanodiscs containing brain lipid extracts, resulting in well-dispersed particles on cryo-EM grids (Supplementary Fig. 1). Yields of production were improved around 2-fold in the α7ΔICDcryo construct as compared α7FLcryo. In parallel, we expressed and purified C4 and E3 nanobodies in a monomeric form as described previously[9] (Supplementary Figs. 1 and 3).

We first verified that both α7cryo constructs retain the modulation properties of C4 and E3 in TEVC (Supplementary Fig. 1). As for the wild-type α7 nAChR, a pre-incubation of 1 μM of E3 leads to the potentiation of ACh-gated currents with no apparent effect on desensitization. C4 pre-incubation has no significant effect by itself but inhibits E3-elicited potentiation (Supplementary Fig. 1). To further document E3 potentiation, we compared it to the effect of PNU-120596. As expected, the effect of PNU on α7FLcryo is different from E3 since it impairs desensitization while E3 has low/no effect on desensitization. We observed an additive effect of PNU and E3 (Supplementary Fig. 1).

We also measured the binding affinity of both nanobodies for the nanodiscs-reconstituted α7ΔICDcryo using single-cycle kinetics Surface Plasma Resonance assays. In both cases, Rmax values suggest a maximum of three nanobodies molecules bound per receptor molecule. Concentration-response experiments show that C4 displays slower association and dissociation kinetics ($k_{on} = 1.2 \pm 0.05 \ 10^6 \ M^{-1}.s^{-1}$, $k_{off} = 0.024 \pm 0.02 \ 10^{-2} s^{-1}$) than E3 ($k_{on} = 11 \pm 0.02 \ 10^6 \ M^{-1}.s^{-1}$, $k_{off} = 1.19 \pm 0.01 \ 10^{-2} s^{-1}$). Data show high affinity for both C4 and E3 with Kd of $0.2 \pm 0.3$ nM and $1 \pm 0.01$ nM, respectively (Supplementary Fig. 4). We also assayed E3 in the presence of nicotine to investigate whether a possible increased affinity could account for its PAM action. However, the affinity of E3 was found identical in the absence and presence of nicotine (Supplementary Fig. 4).

### Cryo-EM structures of α7 in complex with C4 show preservation of nicotine-elicited reorganizations of the ECD

We solved the cryo-EM structure of the α7ΔICDcryo in the presence of a 50-fold excess of C4 (1 μM α7 pentamer for 50 μM C4, C4-Apo structure) (Fig. 1 and Supplementary Fig. 5, Supplementary Table 1). After 2D and 3D classifications, the final set of particles yielded a 5-fold symmetrical reconstruction, with five bound nanobodies per α7 pentamer. For the ECD and nanobodies, the density map is of good quality and allows to confidently build the main and side chains of the protein with a nominal resolution of 2.3 Å. In contrast, the density of the TMD and associated lipid nanodisc, although visible at low contouring of the map, shows poor quality with no elongated and rod-like feature that could be attributed to transmembrane helices (Supplementary Fig. 7). The latch turn, a short helix protruding from the C-terminus of M4 and interacting with the lower part of the cys-loop, is not resolved. This shows that, in MSP1E3D1 nanodiscs containing brain lipid extracts, the TMD conformation of α7ΔICDcryo is heterogeneous among the selected particles, indicating structural flexibility.

In the published structures in nanodiscs[6,7], the variable loop from the ICD was truncated, but the two α-helices MA and MX were kept and resolved, notably the MA helices of each subunit that form a tight quaternary pentameric bundle that could possibly rigidify the TMD. To check if the weak TMD resolution is due to our ICD truncation, we solved the structure of α7FLcryo, also with a 50-fold excess of C4 and added 100 μM nicotine to further stabilize the protein conformation (C4-Nic structure) (Fig. 1 and Supplementary Fig. 6, Supplementary Table 1). The resulting reconstruction reaches a resolution of 3.6 Å for the ECD/nanobody part but also features a disordered transmembrane

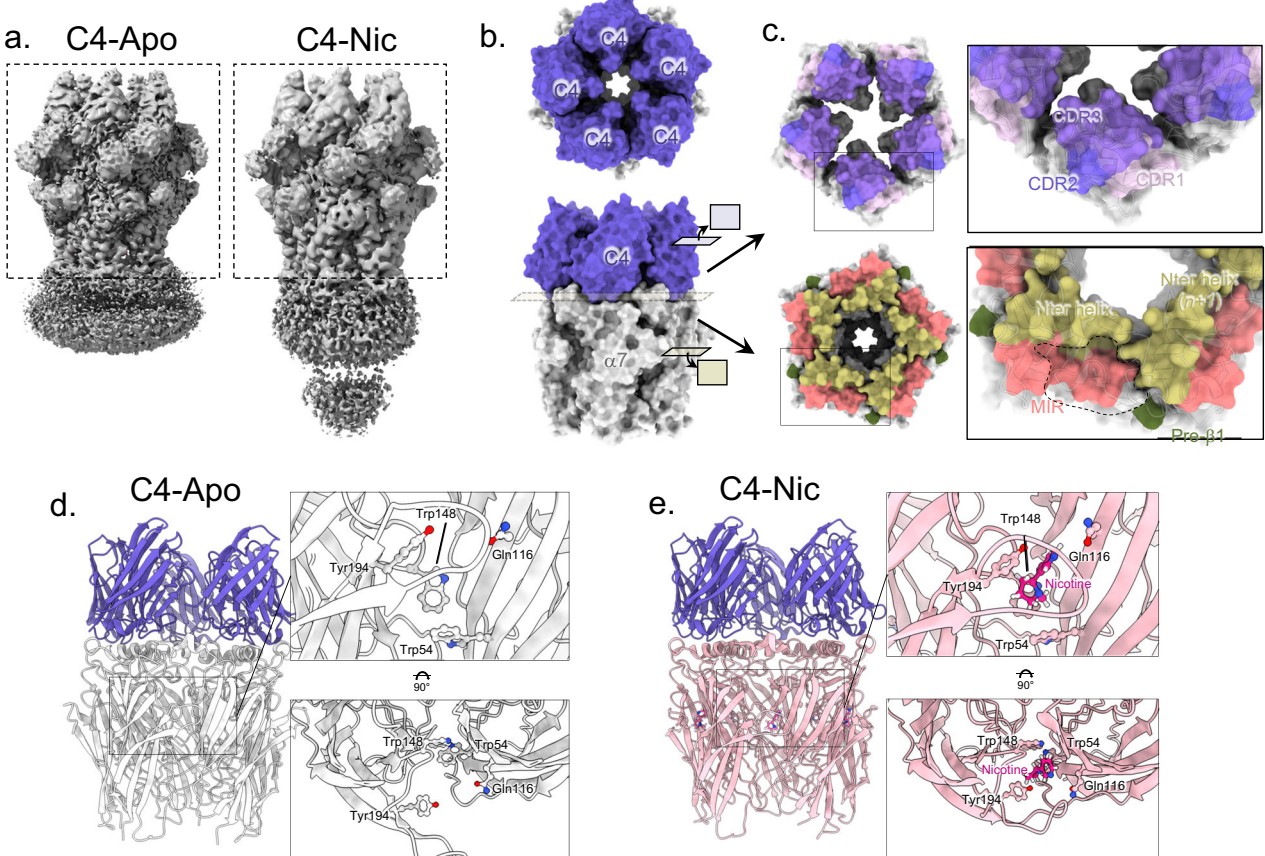

**Fig. 1 | C4-α7 cryo-EM structures in the absence or presence of nicotine.**
**a** Unsharpened maps of the α7ΔICDcryo in complex with C4 (C4-Apo, left) and of the α7FLcryo in complex with C4 and nicotine (C4-Nic, right) with densities contoured at 2σ. The dashed boxes denote the parts that were used for model building. **b** Top and side views of the C4-Apo structure, protein chains are depicted in white (α7 apo-) and purple (C4). The structure is shown exploded in (**c**). **c** Top: Bottom view of the ring of C4 with the molecular surface of the three CDRs colored in shades of purple. Bottom: Top view of the ECD of α7 with N-ter helices, MIR and pre-β1 colored in yellow, pink and green, respectively. The groove in which the CDRs are plunging is delimited with a dashed line. **d** Side view of the C4-Apo structure, represented in cartoon with α7 in white and C4 in purple. The orthosteric binding site is enlarged and further rotated by 90 °C in the right panels. Residues involved in nicotine binding are depicted in sticks. **e** Side view of the C4-Nic structure, represented in cartoon with α7 in pink and C4 in purple. The orthosteric binding site is enlarged and further rotated by 90 °C in the right panels. Residues involved in nicotine binding and nicotine itself are depicted in sticks.

domain, indicating that the flexibility of the TMD is a distinctive feature of the α7 nAChR in our preparations. Still, extra density is visible below the TMD/nanodisc area, compatible with the presence of the bundle of MA helices (Supplementary Fig. 7).

Our data nevertheless allow the model building of the α7 ECD, including the first glycan linked to α7 at Asn23 and Asn67 and two glycans on α7 Asn110, in complex with C4. C4 adopts a classical immunoglobulin-like fold, despite lacking the canonical intra-sandwich disulfide bond (Supplementary Fig. 3). In both C4-Apo and C4-Nic structures, five nanobodies are bound in a similar way and form an additional ring above the ECD. We numbered the nanobody chains according to the α7 subunits anticlockwise arrangement, i.e., C4-chain F (*n*) sits above α7-chainA (principal subunit, *n*), C4-chain G (*n+1*) sits above α7-chain B (complementary subunit, *n+1*). Each nanobody interacts mainly with the top of a principal α7 subunit but also with the α7 complementary subunit and presents its CDRs facing the ECD. The axis of each nanobody β-sandwich is roughly perpendicular to the membrane, in an orientation close to that of the β-sandwiches of the α7 ECDs, which also display an immunoglobulin-type fold. Nanobodies interact substantially with each other (see below), and, while monomeric in solution, they present here a 5-fold assembly. A detailed description of the receptor-nanobody and nanobody-nanobody interactions is provided in sections "Analysis of the E3 and C4 common epitope" and "Inter-nanobody interactions" respectively.

We then analyzed the α7 ECD global conformation by measuring deviations to known structures. First, for the C4-Nic structure, Cα Root-mean-square deviations (rmsd) calculations of the α7 ECD show that the C4-Nic structure aligns best with structures obtained in the presence of agonists, with or without PNU (Fig. 2, Supplementary Fig. 8, Supplementary Table 2). Superimposition of the core of the α7 β-sandwich indeed shows that the whole β-sandwich displays a tertiary/quaternary arrangement that fits near-perfectly the one seen in the "ECD agonist-bound" but not the "ECD apo/Bgt" conformations of the previously solved structures (Supplementary Fig. 8). We quantified this by looking at structural hallmarks conserved in the family (loop C capping, cys-loop motion). Measurements of the Loop C capping (10 Å for the $Cys189_A$-$Tyr117_E$ distance) and of the cys-loop outward motion (11 Å for the $Cys127_A$-$Ile168_E$ distance) clearly show values that are characteristic of active/desensitized rather than resting conformation (Loop C capping around 10 Å vs 17 Å, cys-loop motion around 10 Å vs 8 Å, respectively).

In the five orthosteric sites of C4-Nic, we found an additional density in which we could confidently build a nicotine molecule (Figs. 1 and 2). The pyrrolidine points downward, and its electropositive ammonium lies in a box made up of aromatic residues from loop A (Tyr92), B (Trp148), C (Tyr187, Tyr194) and D (*n+1* Trp54). The pyridine points upward and elicits contacts with loop C (disulfide bridge Cys189-Cys190) nand loop E (*n+1* Leu118). Similar poses of the agonist

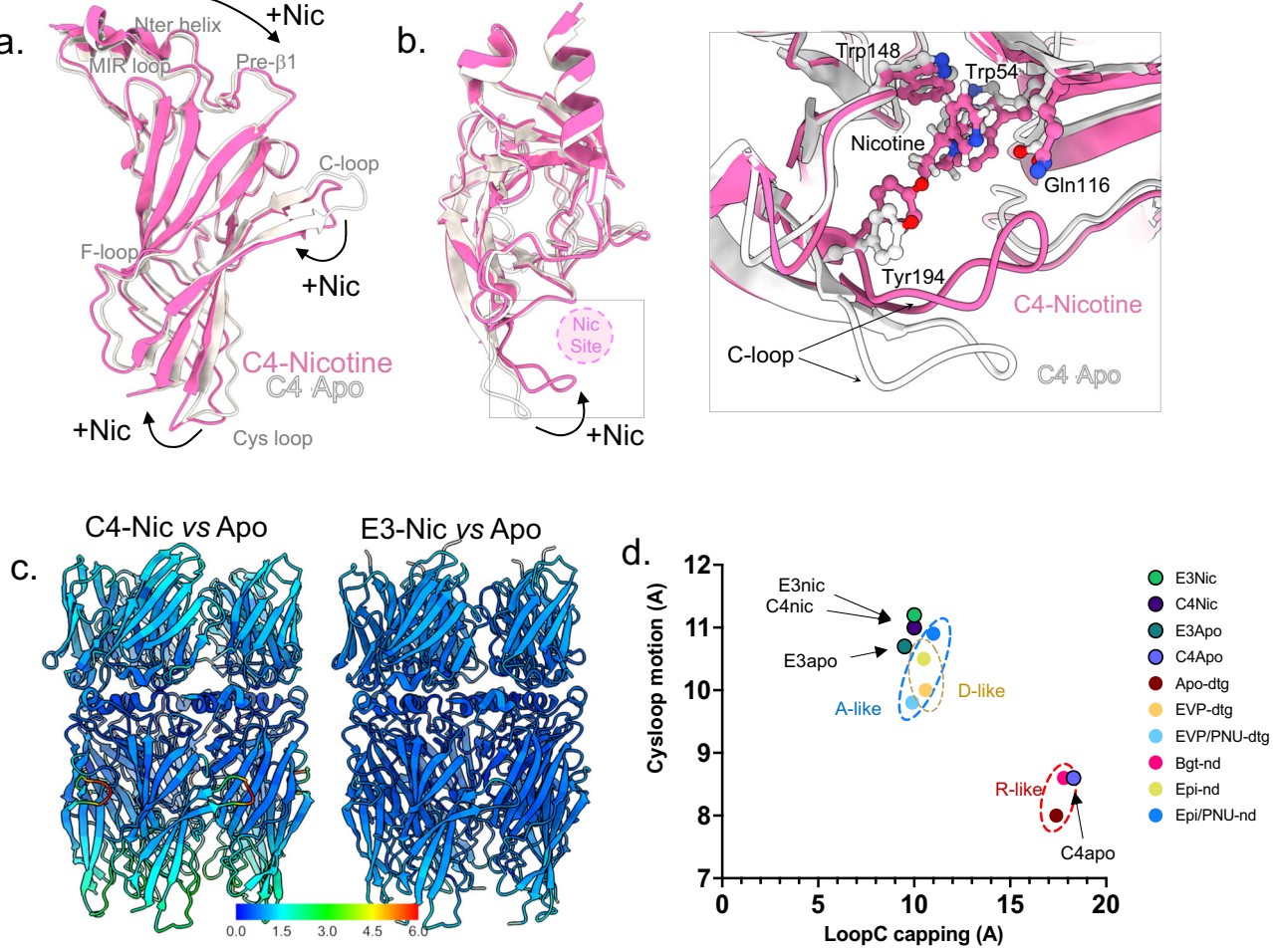

**Fig. 2 | α7 conformation in C4-Apo corresponds to an apo-like ECD conformation, and in C4-Nic, E3-Apo and E3-Nic to an agonist-bound conformation. a** α7 monomers from the C4-apo (gray) and C4-Nic structure (pink) are seen from the side and depicted in cartoons after superimposition of the whole α7 pentamer. Landmark loops are labeled, and motions from Apo- to Nic- are highlighted with black arrows. **b** The same representation as in (**a**) but seen from the top. The nicotine binding site is depicted with a pink circle. A close view of the nicotine binding site is shown on the right. Nicotine and key residues from the binding pocket are labeled and depicted in sticks. **c** Conformational changes upon nicotine binding in C4-bound (left) and E3-bound (right) structures. C4-Nic and E3-Nic are represented in cartoons, and Cα are colored according to the rmsd value (in Å) calculated using their Apo- counterpart, with the same range for both and shown in the color key below. **d** Correlation of the cys-loop outward motion (in Å) and Loop C capping (in Å) in the known structures of α7 allows to define pairs of values that are landmarks of the α7 conformations. R, A, and D-like structures are grouped in dashed circles.

were found in the structure of heteromeric α4β2 and α3β4 nAChRs in complex with nicotine[11,13,14], and in the α7 nAChR in complex with the related alkaloid epibatidine[6]. These observations are consistent with a conformation of the α7 ECD corresponding to an agonist-bound conformation. Given that information about the TMD conformation is missing in our dataset, we are not able to determine whether this conformational change upon nicotine binding is transmitted to the TMD.

For the C4-Apo structure, the α7 ECD structure aligns near-perfectly with that of the Bgt-nanodisc and apo-detergent structures from structure superimposition (Supplementary Fig. 8), rmsd (Supplementary Table 2) and structural hallmarks (Fig. 2 and Supplementary Fig. 8) analysis, with notably an uncapped Loop C (17 Å for the Cys189$_A$-Tyr117$_E$ distance) and no cys-loop outward motion (Cys127$_A$-Ile168$_E$ distance below 9 Å). Those markers indicate that the C4-Apo structure shows an apo-like conformation of α7 ECD. This is best exemplified by the orthosteric binding site, which is empty and displays a wide-open loop C (Fig. 1). In the nanobodies region, a comparison of C4-Apo and C4-Nic structures shows few conformational changes upon nicotine binding apart from a small motion of the apex of the nanobodies away from the pentameric axis (Supplementary Fig. 9).

In conclusion, with five C4 nanobodies bound, the ECD of α7 adopts conformations similar to those observed in the absence of nanobody, namely an apo/Bgt-like conformation in the absence of agonist and an agonist-bound-like conformation in the presence of nicotine. Therefore, C4 appears to be a neutral binder, preserving the nicotine-elicited reorganizations of the ECD in Cryo-EM conditions.

## E3 alone stabilizes an agonist-elicited conformation of the ECD in Cryo-EM structures

We then solved the structure of α7ΔICDcryo with a 50-fold excess of the E3 nanobody, with and without 100 μM nicotine, with an overall resolution of 2.6 and 2.8 Å in the ECD/nanobody region, respectively, and low signal in the TMD (Fig. 3, Supplementary Figs. 10 and 11). Five E3 molecules are bound in a similar arrangement to the one seen for C4. E3 adopts the same overall fold as C4 with local variations.

The E3-Nic structure shows an ECD conformation very close to that of C4-Nic, including the five bound nicotine molecules, consistent with agonist-bound conformations of the ECD (Figs. 2 and 3, Supplementary Figs. 8 and 9). Surprisingly, the conformation of the α7 ECD in the E3-Apo structure also matches best the agonist-bound

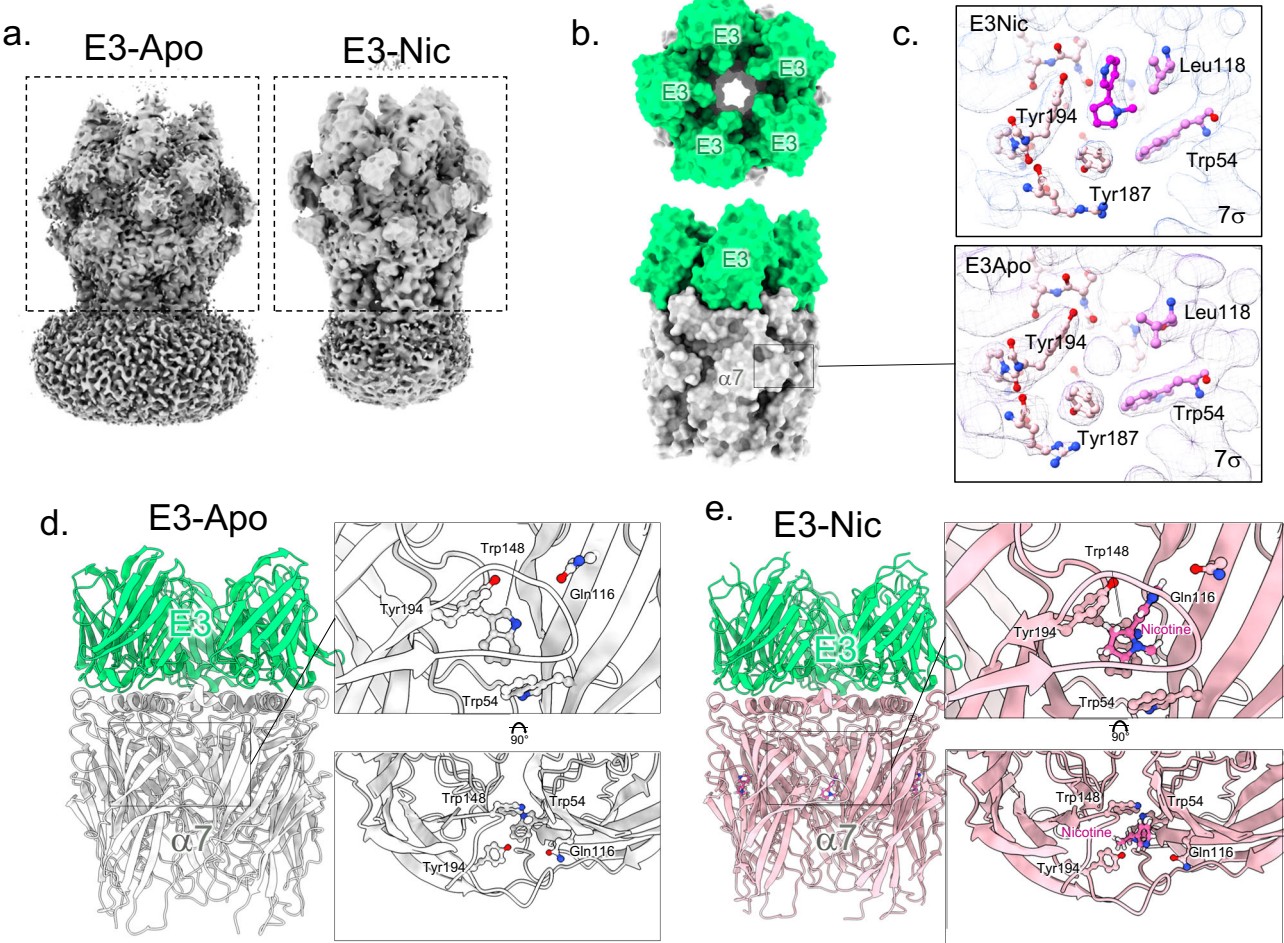

**Fig. 3 | E3-α7 cryo-EM structures in the absence or presence of nicotine.**
**a** Unsharpened maps of the α7ΔICDcryo in complex with E3-Apo (left) and E3-Nic (right) with densities contoured at 2σ. The dashed boxes denote the parts that were used for model building. **b** Top and side views of the E3-Apo structure, protein chains are depicted in white (α7 apo-) and green (E3). The orthosteric sites are shown in (**c**). **c** Neurotransmitter binding pocket of E3-Nic and E3-Apo. Electron densities are contoured at 7σ, and residues involved in neurotransmitter binding are shown in sticks. Nicotine was built in the extra density found in E3Nic, while the small and spherical density found in E3-Apo likely fits a cation or a water molecule. **d** Side view of the E3-Apo structure, represented in cartoon with α7 in white and E3 in green. The orthosteric binding site is enlarged and further rotated by 90 °C in the right panels. Residues involved in nicotine binding are depicted in sticks. **e** Side view of the E3-Nic structure, represented in cartoon with α7 in pink and E3 in green. The orthosteric binding site is enlarged and further rotated by 90 °C in the right panels. Residues involved in nicotine binding and nicotine itself are depicted in sticks.

conformations, including Loop C capping and Cys-loop outward motion (Figs. 2 and 3). At the orthosteric site, the conformations of all binding determinants resemble the ones seen in the E3-Nic structure. In the cavity, we found a well-defined small spherical density, which is located where the electropositive pyrrolidine ammonium would be. The density would be compatible with water, or more likely, a cation such as $Na^+$ that may contribute, along with the binding of E3, to compact the orthosteric site in the absence of nicotine (Fig. 3). Yet, Cα rmsd calculations show that E3-Apo slightly differs from both E3-Nic and C4-Nic, with locally 1–2 Å deviations in the loop C, as well as the top and bottom regions of the ECD (Supplementary Fig. 9). In addition, we could observe a slight expansion of the ECD subunit in E3-Nic vs E3-Apo (~1 Å displacement), a motion comparable with the one observed between the Epi vs Epi/PNU structures in nanodiscs (PDB 7KOQ and 7KOX, Supplementary Fig. 9) but absent between the EVP and EVP/PNU structures in detergent (PDB 7EKP and 7EKT). As seen with C4, nicotine binding triggers a small motion of the tip of the nanobody away from the channel axis (Supplementary Fig. 9).

In conclusion, our data point to a striking effect of E3, which, in the cryo-EM conditions (high 50 μM E3 concentration, 1 h equilibration period before freezing in the grids), stabilizes by itself an agonist-bound conformation of the ECD through long-range allosteric interactions at the top of the receptor.

## Analysis of the E3 and C4 common epitope
C4 and E3 bind to α7 in a highly similar manner (Fig. 4), their CDRs contacting the top-ECD platform formed by the principal α7 subunit down below (with a ~500 Å² surface of interaction), and the complementary α7 subunit (~300 Å²). On the principal subunit, it is formed by the C-terminal half of the top α helix that faces the vestibule (N-ter helix) and by the long β2-β3 loop, also called the Main Immunogenic Region (MIR)[15], on the outer side. On the complementary subunit, it is formed by the N-terminal half of the top α-helix (N-ter helix).

Both nanobodies display unique disulfide bond patterns. First, E3 harbors the "canonical" disulfide bond that links the strands that just precede the CDR1 and CDR3, which is missing in C4. Second, the CDR3 of E3 is longer by two residues and is bent upward by the presence of an extra disulfide bond between the end of the CDR3 and the second β-strand of the framework region 3 (FR3) (Supplementary Fig. 3). In known nanobodies featuring a long CDR3, a supplementary disulfide bond rigidifying the structure is common[16]. For both nanobodies, conserved side chains from the three CDRs ($His32_{CDR1}$, $Trp54_{CDR2}$,

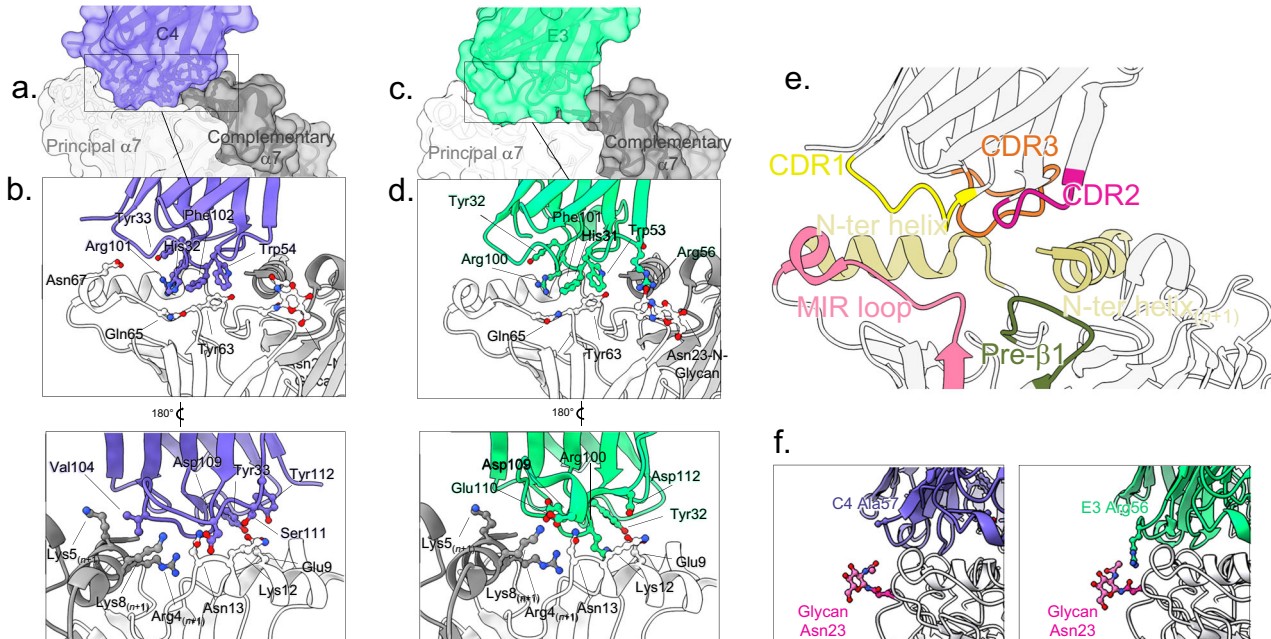

**Fig. 4 | Binding pose and epitope of C4 and E3. a** Overview of the binding site of C4 on α7 exemplified by the C4-Apo structure. One C4 molecule (purple surface and cartoon) contacts two α7 subunits depicted in light (principal n subunit) and dark (complementary n+1 subunit) surface and cartoons. **b** Details of the binding site as seen from the solvent (upper panel) or the vestibule (lower panel) of α7 with color code as in (**a**). key residues are labeled, and their side chains displayed in stick balls. **c** Same as in (**a**) but with the E3-Apo structure, E3 is colored in green. **d** Same

as in (**b**). but with the E3-Apo structure, E3 is colored in green. **e** Overview of the loops involved in binding exemplified by the C4-Apo structure represented in cartoons. **f** Interaction of E3 with Asn23 glycan. Close view of the Glycan (pink sticks) built on α7-Asn23 (pink sticks) from the pre-β1 of α7 (gray) and of the CDR2 of C4 (left, purple) or E3 (right, green). Ala57 from C4 and its homologous Arg56 from E3 are depicted in sticks.

Arg101$_{CDR3}$ and Phe102$_{CDR3}$, C4 numbering) are directly plunging into a groove bordered by the helix and MIR elements. Both nanobodies elicit a complex pattern of interactions with α7 that is detailed in Supplementary Tables 3 and 4. Although the sets of interactions are different in C3 and E4, they involve conserved residues from the CDR1, CDR2 and first half of CDR3 (Gly27$_{CDR1}$, Tyr33$_{CDR1}$, Trp54$_{CDR2}$, Arg101$_{CDR3}$, Phe102 $_{CDR3}$, C4 numbering). They involve also non-conserved residues from the second half of the CDR3, where C4 residues Asp109 and Ser111 are engaged in an extensive set of interactions with the principal N-terminal helix, while much fewer contacts are found in E3 (Fig. 4, Supplementary Tables 3 and 4). In addition, E3 shows a unique interaction between the Arg56$_{CDR2}$ guanidinium and both the main chain carbonyl and the first glycan grafted on α7Asn23 (pre-β1), an interaction absent in C4 that harbors an alanine in place of the arginine at this position (Fig. 3). Overall, the CDR3 is the main interacting loop, contacting the MIR and the N-terminal helices of both subunits.

We further investigated the contribution of specific α7 motifs to nanobody binding by mutagenesis of α7FLcryo. On the one hand, we mutated the charged residues of the N-terminal helix of α7 (R4Q K5Q K9Q triple mutant and E9Q K12Q N13A triple mutant), and on the other hand, we replaced the MIR residues with the aligned MIR of the α3-nAChR which does not bind C4 and E3[9]. Finally, we prevented N-linked glycosylation by mutating the serine of the Asn-X-Ser motif (Ser25Ala). We performed immunofluorescence assays in non-permeabilized HEK293 cells (Supplementary Fig. 4). Immunolabelling of the extracellular C-terminal Rho1D4 tag shows that the MIR and glycosylation mutants, but not the N-terminal helix mutants are expressed at the cell surface. Immunolabelling with E3 and C4, using nanobodies fused to the heavy chain fragment of a human IgG (constructs detailed in ref. 9, Supplementary Fig. 4), shows clear labeling of cells expressing the WT and glycosylation mutant, but no labeling of the MIR mutant. This confirms that the MIR is essential for the nanobody binding, while the glycosylation on Asn23 is not mandatory for E3 high-affinity binding.

### Inter-nanobody interactions

A striking feature of the structures of α7 bound to C4 and E3 is the 5-fold assembly adopted by the nanobodies above the receptors (Figs. 1 and 3), where the FR3 region of each C4/E3 interacts with the FR1 region of the n+1 C4/E3 nanobody through a ~200 Å² interface (Supplementary Fig. 8). In C4, Tyr60, Ser63 and Lys 66 contact n+1 Gln2 and Gln4 (Fig. 5) and in E3, Asn62 and Lys65 contact n+1 Gln3 and Gln5 and Gln 116. The CDR3 of E3 also contributes to the interface, the guanidinium of Arg108 lying near the carboxylate of Asp 113, suggesting a salt-bridge interaction. A more extensive set of inter-nanobody interactions is thus observed in E3 vs C4.

To further explore the quaternary binding mode of the nanobodies, we solved the structure of α7ΔICDcryo in the presence of a sub-saturating concentration of C4 (10 μM) (Fig. 5 and Supplementary Fig. 12; Supplementary Table 1). Indeed, in the early stages of the cryo-EM work at the 2D classification step, we noticed that, for both E3 and C4, the addition of 10 μM of nanobodies is not sufficient to reach the C5-symmetry in the nanobody regions, indicating partial nanobody binding (Supplementary Fig. 13). We processed a dataset obtained with 10 μM of C4 and no nicotine. 2D class averages show a minor fraction of particles containing one or two bound nanobodies and a major fraction containing three bound nanobodies. We selected these later particles to solve a C4partial-Apo structure. Without imposed symmetry, we obtained a final volume of 4.4 Å resolution and, after map improvement using the density modification tool of Phenix[17], we were able to build a model where the main chains and most of the side chains of the α7 ECD and the three nanobody molecules are confidently built. For simplicity, we labeled the three C4 according to their position in C4-Apo: C4 chains F, G and I sit above α7 chains A, B and D, respectively. Interaction determinants of the three C4s with α7 are identical to the ones observed in C4-Apo. C4$_F$ and C4$_G$ interact with each other in a similar manner to C4-Apo, although through a slightly larger interaction surface (~260 Å², Supplementary Fig. 8), while C4$_I$ sits alone.

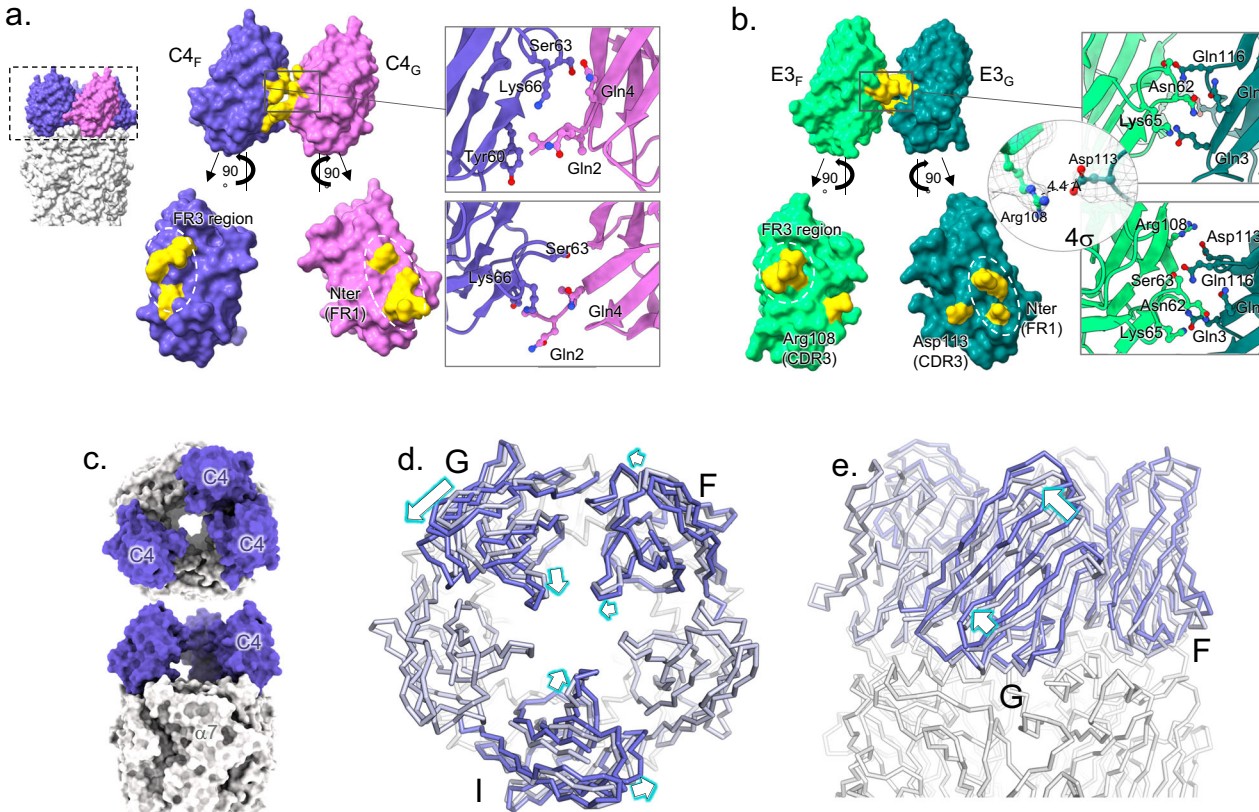

**Fig. 5 | Nanobody-nanobody interactions. a** Surface representation of two adjacent C4 (F in purple, G in pink) in C4-Apo and their exploded view below. Residues at the interface are colored in yellow, and their details are shown in cartoons and sticks in a close-up view (right). **b** Surface representation of two adjacent E3 (F in light green, G in dark pink) in E3-Apo and their exploded view below. Residues at the interface are colored in yellow, and their details are shown in cartoons and sticks in a close-up view (right). Additionally, the putative salt bond between Arg108 and Asp113 is detailed in a separate close-up view, with the density contoured at 4σ. **c** Overview of the C4partial-Apo structure from the top (top) and the side (bottom)

with α7 in gray and the three C4 molecules in purple. **d** Superimposition of the C4partial-Apo on the C4-Apo structures, aligned on α7 ECD. Both structures are represented in ribbon, α7 in gray, C4 from C4partial-Apo in purple and C4 from C4-Apo in gray blue. Motion from C4-Apo to C4partial-Apo are highlighted with arrows. On the left, structures are seen from the top and show a lateral inclination of the nanobodies toward the anticlockwise subunit accompanied by a radial inclination toward the vestibule. **e** Same as in (**d**), but with structures seen from the side, with the most displaced nanobody in front (chain G).

Interestingly, each C4 chain displays a different radial and lateral inclination. The F, G and I chains are more inclined toward the vestibule (radial angles of 129.3°, 128.8° and 126.0°, respectively), compared to C4-Apo and C4-Nic (133.4° and 133.8°) (Supplementary Fig. 13). Likewise, the F, G and I chains are more inclined toward the (n+1) adjacent subunit (lateral angles of 82.8°, 80.5° and 78.2°, respectively), compared to C4-Apo and C4-Nic (85.1° and 85.3°). Therefore, data show that the nanobodies can bind independently from each other but that their orientation is constrained by the presence of neighboring nanobodies. This suggests a sequential binding mode, where a single bound nanobody adopts a marked inclination toward the vestibule and the (n+1) adjacent α7 subunit, where two adjacent nanobodies show intermediate inclinations and five bound nanobodies adopt a straighter binding orientation. This later is necessary for the α7 receptor to accommodate five bound nanobodies without steric clashes, eventually generating a symmetrical bundle (Fig. 5).

### E3 has a direct-agonist effect on α7 L247T mutant

Our Cryo-EM data suggest that E3 could stabilize a global agonist-bound conformation at the plasma membrane. In functional assays, E3 binding would thus be predicted to increase the apparent affinity of α7 for its agonists. To investigate this possibility, we performed ACh dose-response curve with or without pre-incubation of 1 μM E3 (15" before each ACh test) in TEVC. As previously reported[9], the apparent affinity for ACh is not significantly affected by E3 (Supplementary Fig. 14).

However, we noticed that at high ACh concentration, where desensitization often decreases the current peak in oocyte TEVC, the potentiating effect of E3 is absent. We hypothesized that desensitization would prevent visible potentiation from E3 at those ACh doses.

To challenge this idea, we used the α7 mutant L247T that disrupts a central hydrophobic ring in the channel[18,19] (Fig. 6). This mutant strongly reduces desensitization and produces a marked gain of function phenotype, increasing the efficacy of partial agonists. For this mutant, 15" 1 μM pre-application of E3 potentiates currents at most ACh concentrations, yielding a significant increase in ACh apparent affinity in the presence of E3 (17 ± 20 μM versus 4 ± 1 μM with E3, n = 10, p < 0.05 in an unpaired t-test). In conclusion, the oocyte TEVC system makes it difficult to determine the precise changes in ACh affinity on the wild-type α7, but data obtained on the L247T indicates that E3 significantly increases the apparent affinity of the receptor for ACh.

Furthermore, our Cryo-EM data suggest that high E3 concentrations and long incubation periods could trigger activation and/or desensitization in the absence of an orthosteric agonist. Those conditions are hardly reachable in a standard setup and with the WT, and 1 μM E3 does not trigger activation. However, while performing the ACh dose-response curves on the L247T mutant, we observed an increase of the holding current during the pre-application of 1 μM E3 that represents 6.0 ± 1.8 % of the response to 0.3 μM ACh (Fig. 6). This shows that E3 does have significant agonist properties in the absence

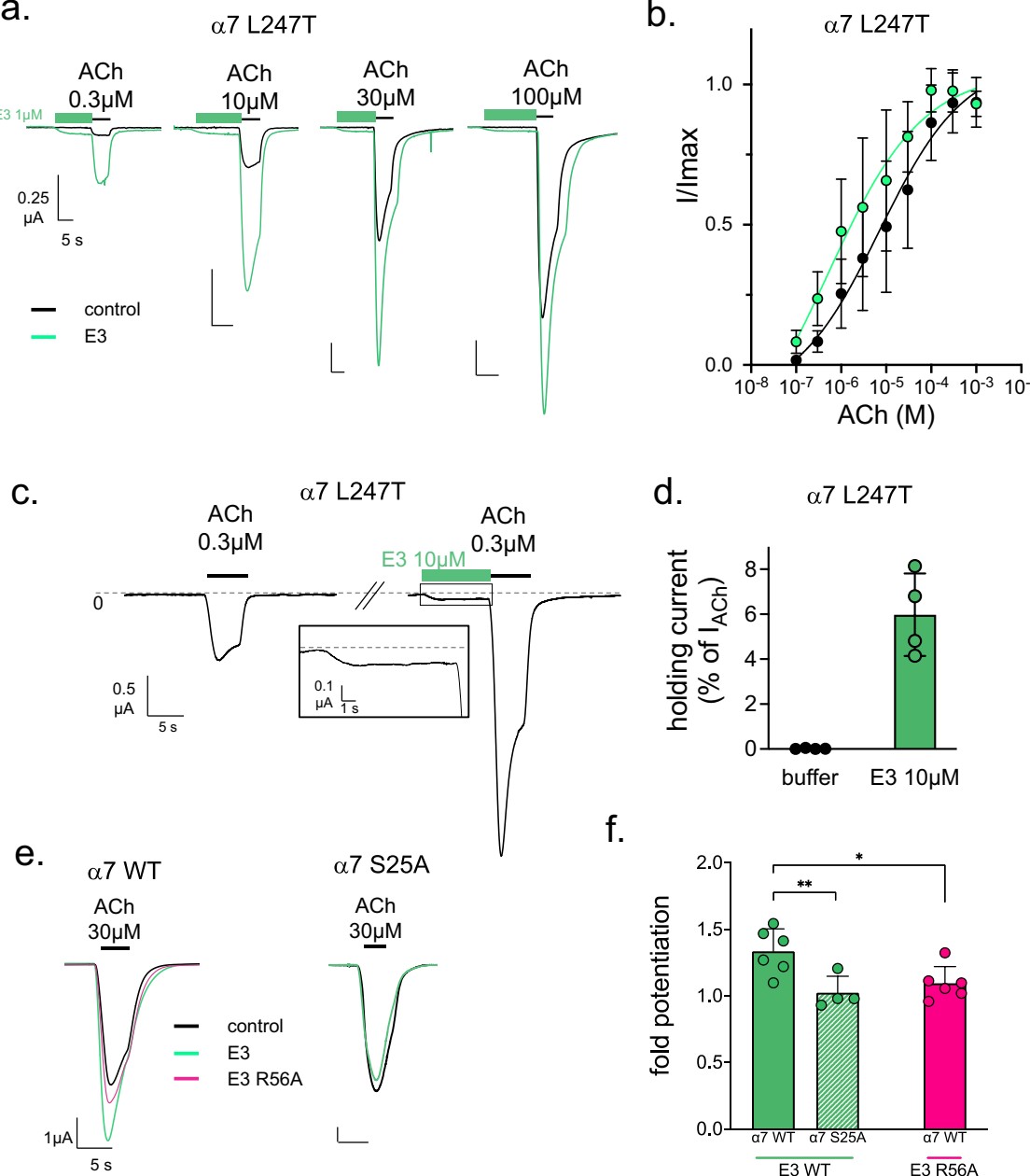

**Fig. 6 | Electrophysiological analysis of the potentiation by E3. a** Representative traces of ACh dose-response curves of the α7 L247T mutant with (green) or without (black) 15-s pre-application of 1 μM E3 on a single cell. **b** Resulting dose-response curves after normalization on maximal currents of both conditions. Points are mean ± s.d. with *n* = 11 cells. **c** E3 direct activation on α7 L247T mutant recorded by TEVC on *Xenopus* oocytes. Representative trace showing a first 0.3 μM ACh application yielding slow-desensitizing currents. After 2-min washing, 10 μM E3 was applied for 10 s, evoking significant current, before 0.3 μM ACh. A close-up view of the recording during E3 application is shown in a black box. **d** Holding currents

measured on α7 L247T with or without 10 μM E3 normalized with the non-potentiated response to 0.3 μM ACh. Points are mean ± s.d. with *n* = 4 cells. **e** Potentiation assays of E3-WT and E3-R56A on α7 WT and of E3-WT on α7-S25A. Responses to 30 μM ACh in the absence of E3 (black) are superimposed with response after 30 s application of 1 μM of E3 wild-type (green) or R56A (pink). **f** Fold potentiation calculated on *n* = 4 to 6 cells by E3-WT on α7 WT and S25A mutant and of E3-R56A on α7-WT. Bars are mean ± s.d. Values were submitted to an unpaired *t*-test with *$*p$ = 0.0183, *$**p$ = 0.0013.

of ACh when applied for a short period at a relatively high concentration on a weakly-desensitizing mutant.

### E3-elicited potentiation involves interaction with Asn23-linked glycan

The cryo-EM structures show a sticking allosteric action of E3, a feature that parallels its PAM activity in electrophysiological experiments. In contrast, C4 appears "silent" with both techniques. We thus investigated whether specific E3-α7 contacts are involved in the PAM activity.

To this aim, we compared the E3 and C4 structures, searching for specific motifs/interactions of E3 absent in C4, followed by mutational and functional assays.

First, to investigate a possible contribution of the CDR3 conformation of E3, we removed the extra disulfide bond in the C95S/C114V E3 mutant, but it displays an unaltered potentiating effect in TEVC (Supplementary Fig. 4).

Second, a key feature of both nanobodies is their binding to the *n*+1 nanobody and α7 subunit. Since α7 undergoes a substantial

quaternary reorganization during gating, the interaction of E3 with the complementary chains could contribute to constrain the relative arrangement of two adjacent α7 ECDs toward an active/desensitized conformation. We mutated specific potential salt bridges between E3 and E3($n$+1) (Arg108-Asp113), as well as between E3 and α7($n$+1) (Glu110-Lys8$_{α7n+1}$). E3-R108Q and E3-E110Q show an intact affinity by SPR and a slightly weaker, yet not significantly different from E3, PAM activity by TEVC (Fig. 6 and Supplementary Fig. 4).

Third, a unique feature of E3 is the interaction between Arg56$_{CDR2}$ and the first sugar moiety linked to Asn23 from the principal subunit. We showed in mutating α7S25A that the glycosylation is not mandatory for E3 high-affinity binding. Interestingly, α7S25A is no more potentiated by E3, suggesting a key contribution of the first sugar moiety in potentiation. However, we cannot exclude that the mutation, although qualitatively showing binding in immunofluorescence, could decrease the affinity for E3 to such an extent that weak binding would occur in our electrophysiological assay (1 μM E3). To address this issue, we tested the E3-R56A mutant. It shows an affinity identical to that of E3-WT in SPR (Supplementary Fig. 4) and almost totally loses its ability to potentiate ACh-gated currents in electrophysiology (Fig. 6 and Supplementary Fig. 4).

In conclusion, data pinpoint an Arg-sugar interaction at the E3-α7(n) interface which mutation strongly decreases the potentiation, and two putative salt-bridges at the E3-α7($n$+1) and E3-E3 interfaces whose mutation show a tendency to decrease the potentiation.

## Discussion

In this study, we solved five structures of the α7-nAChR in complex with two nanobodies, unraveling a common epitope at the apex of the receptor, which is formed by the MIR and the N-terminal helices from two adjacent subunits. In the pLGIC family, nanobodies were previously generated and used mainly to assist structural determination. Most bind at the ortho- or pseudo-ortho-steric site in the 5HT$_3$R[20], GABA$_A$R[21,22] and prokaryotic homolog ELIC[23], except for a single NAM nanobody binding in the vestibule of ELIC[23]. It is noteworthy that E3 and C4 were generated through immunization of alpacas with cells expressing the fully glycosylated α7-nAChR ECD, while previous nanobodies involved immunization with purified proteins partly depleted from N-linked glycosylation by enzymatic cleavage (5HT$_3$R) or production in GntI- cells (GABA$_A$R). Interestingly, the unsharpened map of the E3/C4-bound structures suggests that glycosylation trees could substantially mask the orthosteric site (Supplementary Fig. 7), thereby plausibly preventing the generation of binders targeting this site and favoring a more apical epitope. Of note, the epitope of C4 and E3 shows weak sequence conservation among the various nAChR subtypes, accounting for their high specificity to the α7-nAChR with no observed binding by immunofluorescence to the α4β2 and α3β4 nAChRs[9].

Several Fab fragments of conventional IgG were also found to bind within or around the orthosteric and pseudo-orthosteric sites in GABA$_A$R[24,25] and α4β2 nAChR[13]. More importantly, the MIR was first identified in the α1 subunit of the muscle-type nAChR as a key epitope for autoantibodies causing *myasthenia gravis*[15,26]. The crystal structure of the monomeric ECD of α1 in complex with a Fab fragment of one of those autoantibodies shows a binding mode involving the interaction of both Fab chains with the MIR and the N-terminal helix of α1[27]. A somewhat similar binding mode is seen for an anti-α3 Fab in complex with the α3β4 nAChR[14]. However, both structures show a pose of the antibodies tilted to the edge of the top-ECD platform, the epitope centered on the MIR rather than the groove formed between the N-ter helices and the MIR, as seen with E3 and C4. Therefore, the epitope of E3 and C4, while containing the MIR region, is unique among those so far characterized in pLGICs.

A unique feature of C4- and E3-bound structures is also the "pentamerization" of the nanobodies when binding to the α7 template.

The C4partial-Apo structure shows that C4 binds in a sequential manner, a process requiring significant reorganization (change of radial and lateral inclination) of already bound nanobodies to accommodate new ones. Interestingly, SPR shows that C4 displays 0.1 nM affinity with an evaluated maximal occupation of 3 nanobodies per pentamer. In contrast, Cryo-EM experiments show that 10 μM of C4 is not sufficient to saturate the receptor, with full saturation requiring a 50 μM concentration. Data thus suggest that, within α7 pentamers, the first binding events display high affinity (nanomolar from SPR) while the last binding events would display micromolar affinity. We suggest that the sequential binding mode, where the binding of nanobodies is impaired when adjacent positions are already occupied, might contribute to this negative cooperativity. This mechanism might also apply to E3, which displays similar differences in SPR vs Cryo-EM nanobody binding features.

Cryo-EM and TEVC experiments provide compelling evidence for a strong allosteric effect of E3 but not of C4. One limitation of our cryo-EM data is the high flexibility of the TMD that precludes model building. In the different structures, we thus cannot determine the state of the channel that would allow functional annotation. Yet, we show that the ECD displays intact conformational rearrangements upon agonist binding, allowing the mechanistic study of the nanobody that binds to the ECD away from the TMD. In addition, we show that E3 alone stabilizes a conformation of the ECD that is characteristic of agonist-bound structures. This nicely parallels the electrophysiological data showing that E3 acts both as a PAM and a direct partial agonist on the gain of function a7L247T mutant. We propose that the stabilization of the agonist-bound conformation of the ECD by E3 drives its potentiation in functional studies. To further support this idea, the removal of the sugar linked to α7 Asn23, which contributes to the E3 but not to the C4 epitope, indeed strongly impairs the PAM activity of E3 by TEVC.

The PAM activity of E3, evaluated by potentiation of current elicited by 30 μM of ACh, is maximal above 1 μM with an apparent EC$_{50}$ around 100 nM. SPR measures an affinity of 1 nM, suggesting that several binding events might be required to detect a PAM activity. In contrast, low μM concentrations of E3 are not sufficient to detect a direct agonist action. We show that 10 μM E3 alone weakly but significantly activates the α7L247T mutant and increases its apparent affinity for ACh and that 50 μM E3 alone drives α7 ECD into an agonist-bound conformation in cryo-grids. Direct activation by E3 on a7L247T mutant is thus likely elicited by a near-complete or complete saturation of the binding sites on the α7 pentamer.

E3 mediates allosteric effects through its binding at the top platform of the receptor. Beside the orthosteric site, the α7 nAChR is known to carry allosteric sites at the TMD, binding notably ivermectin[28] and PNU[5,7], at the ECD-TMD interface where calcium binds[29], and at a vestibular site located nearby the orthosteric site, that binds small fragments[30]. In the pLGIC family, another allosteric site is well established, the binding site of AM-3607, a PAM binding between the top platform and the orthosteric site of the α3GlyR[31]. Thus, the allosteric site targeted by E3 on α7 is unique and involves a large surface involving multiple contacts. E3 is strongly anchored in the principal α7 subunit down below, with bulky side chains plunging into a groove bordered by the MIR and the helix. The PAM activity is strongly impaired by disrupting a key polar interaction with a sugar from the principal subunit near the interface with the complementary subunit and possibly two putative salt bridges with the complementary $n$+1 E3 and α7 subunit. We speculate that the binding of E3 constrains the relative arrangement of the principal and complementary subunits of α7, favoring a subunit tilt associated with gating. Therefore, E3 implements a novel allosteric site which is the most remote from the core of the coupling pathway between the orthosteric site and the activation gate of the channel.

In conclusion, our work reveals a unique mechanism where nanobodies prolong the length of the pentameric architecture of the ECD to control the global allosteric conformation of the receptor. The modular architecture of pLGICs, made up of ECD and TMD domains that fold in a partially autonomous way[32–34], is thus artificially extended here by an extra-module that tightly binds and is allosterically coupled to the native receptor. Such regulation by additional modules is reminiscent of receptors carrying a regulatory N-terminal domain, such as the pLGIC Declic[35] or the tetrameric glutamate receptors. Finally, from a pharmacological perspective, E3 constitutes an original and highly subtype-specific therapeutic antibody candidate for α7-linked pathologies, including autoimmune[36], neurodegenerative or psychiatric diseases.

## Methods

### Lentiviral vectors construction and production
The lentiviral transfer plasmid was generated from the pHR-CMV-TetO2-IRES-EmGFP and pHR-CMV-TetO2-IRES-mRuby2 plasmids available on Addgene. Human cDNA for α7-nAChRs and NACHO inserts were cloned using an In-fusion cloning kit (Takara). After cloning in the lentiviral transfer plasmids, genes are preceded by a CMV promoter and a tet repressor element (CMV-TetO2) and followed by an IRES element promoting the expression of GFP (IRES-GFP, α7) or mRuby (IRES-Ruby, Nacho).

Viral particles were generated by co-transfection of HEK-293T cells by the lentiviral transfer plasmid, a packaging plasmid and an envelope plasmid. Two days after transfection, viral particles were harvested in the supernatant, treated with DNaseI, filtered through 0.45-µm pores, concentrated by ultracentrifugation and resuspended in PBS. Viral stocks were stored in small aliquots at −80 °C before use. Viral titers were estimated by quantification of the p24 capsid protein using HIV-1 p24 antigen immunoassay (ZeptoMetrix) and by fluorescence-activated cell sorting (FACS).

### Lentiviral transduction, inducible expression in Trex cells
A mixture of α7-containing and NACHO-containing lentiviral particles (10:1) was used to transduce HEK T-Rex 293 cells (Thermo Fisher). The resulting inducible cell line was cultured in suspension flasks in Freestyle 293 expression medium supplemented with blasticidin (5–10 µg/ml final) and FBS (1%) in an orbital incubator at 37 °C and 8% $CO_2$. Protein expression was induced at a density of $3-4 \times 10^6$ cells/ml by the addition of doxycycline (10 µg/L final). The cells were further cultured for 48 h before harvesting by centrifugation.

### α7-nAChR purification
After low-speed centrifugation, the pellet was washed with cold PBS 1x and centrifuged again. Cells were resuspended in buffer A (20 mM Tris pH8.0, 250 mM NaCl) supplemented with an antiprotease cocktail and mechanically lysed (Ultra-turrax T20, IKA, 9 × 30 s). Lysed cells were centrifuged at low speed (2000×g) to remove debris, and intact cells and membranes were collected by ultracentrifugation (190,000×g for 2 h). All further steps were carried out at 4 °C. Membranes were mechanically broken and were resuspended in buffer A (10 ml of buffer per gram of membrane) and supplemented with 1% w/v final of dodecyl-maltoside (DDM, Anatrace) for 1 h solubilization under gentle stirring. The insoluble material was removed by ultracentrifugation (190,000×g for 45 min). The supernatant containing solubilized proteins was bound overnight at 4 °C on a gravity flow Rho1D4 resin (Cube Biotech) equilibrated with buffer A supplemented with 0.05% DDM. After binding overnight, the resin was washed with buffer A and 0.05% DDM and gradually washed to remove DDM and exchange it for 0.01 % Lauryl Maltose Neopentyl Glycol (LMNG, Anatrace).

MSP1E3D1-Histag plasmid was obtained from Addgene, and protein was purified as described in Ritchie et al.[37]. Porcine Brain Lipids Total extracts (BLT, Avanti) were resuspended in buffer A containing 0.01% LMNG at 10 mM, sonicated and stored at −80 °C until use.

The α7 receptor on its resin was incubated with MSP1E3D1 and BLT extracts (ratio 1:2:200) for 1 h at 4 °C. The removal of detergent was initiated by the addition of bio-beads (BioRad) twice for 1 h at 4 °C. The receptor-lipidic disc mixture was washed in buffer without DDM and was eluted with 500 µM Rho1D4 peptide (Cube biotech) for 4 × 1 h at 4 °C. The receptor-lipidic disc mixture was concentrated to 300–400 µl. Polishing was done with a binding on a cobalt resin for 15 min at 4 °C, washed with buffer A and was eluted with 250 mM of imidazole. The fractions were pooled and concentrated to 1–5 µM. Imidazole was removed by washing at the concentration step.

### Nanobodies purification
Genes coding for C4 and E3 nanobodies (and E3 mutants) with cMyc tag and 6xHis-tag at the nanobody's C-terminal were cloned into a pFUSE-derived vector (InvivoGen). The vector was used to transform Expi293F mammalian cells (Thermo Fisher), and protein expression was carried out according to the manufacturer's recommendations using 100 ml final volume of cell suspension. Protein was then purified from the expression medium by affinity chromatography on a 1 ml HiTrap TALON crude column (Cytiva). After sample application, the column was first washed with 10 column volumes of PBS and 10 column volumes of PBS supplemented with 10 mM Imidazole (pH = 7.4) (Sigma); the protein was subsequently eluted with 5 column volumes of PBS supplemented with 0.5 M Imidazole (pH = 7.4). Affinity-eluted nanobodies were finally polished on a HiLoad 16/600 Superdex 200 pg Pre-packed column (Cytiva) using PBS buffer.

### Cryo-EM sample preparation
Typical grids were prepared with 1 µM α7 pentamer supplemented with 10 or 50 µM of nanobodies and 100 µM nicotine when applicable. For preparation of cryo-EM grids, UltrAuFoil grids 300 mesh 1.2/1.3 or 0.6/1.0 from Quantifoil were glow discharged in a PELCO easiGlow for 25 mA/15 s. Three microliters of protein were applied to grids before blotting for 6 s at 100% humidity and 10 °C. Grids were plunge-frozen with a Vitrobot Mark-IV (Thermo Fisher). Frozen grids were stored in liquid nitrogen until use.

### Data collection, processing and model building
Datasets were collected on a ThermoFisher Glacios equipped with a Falcon 4i detector at Institut Pasteur, or a ThermoFisher Titan equipped with a Gatan energy filter bioquantum/K3 at Institut Pasteur or on a ThermoFisher IC-Krios equipped with a SelectrisX and a Falcon4i detector at EMBL-Heidelberg. All datasets were acquired in counting mode using EPU (ThermoFisher) or SerialEM (see Supplementary Table 1 and Supplementary Figs. S4, S5, S9–11 for datasets specific details). The beam-induced motion was corrected by MotionCor2, and defocus values were estimated by CTFFIND-4.1. Laplacian-of-Gaussian algorithm from Relion4 or blob-picker from CryoSPARC3.3.2 were used to pick particles that were submitted to 2D classification. 2D classes showing landmark secondary structures of α7-nAChR were used to pick again particles. After two rounds of 2D classification, classes with clear secondary structures characteristic of α7 were selected as the initial pool of particles used for further processing. Unsymmetrized initial model was built in Relion4, symmetrized in C5 (or unsymmetrized in the case of C4partial) and used for Relion4 unsupervised 3D classification with a 60 Å initial low-pass filter. When indicated, an additional 3D classification without alignment was performed. The final pool of particles was used for 3D refinement in Relion4 with C5-symmetry imposed and relaxed in C1 to account for local heterogeneity, using a mask that includes the signal we assigned to the transmembrane domain. FSC plots were calculated using the post-processing tool from Relion4.

Maps from the refinement runs were sharpened in Phenix1.20, except for the C4partial-Apo volume that was subjected to Phenix1.20 density modification using an initial model made with C4-Apo depleted in two nanobodies molecules. Low sigma value allows to visualize the transmembrane and intracellular regions, but they disappear at sigma values compatible with model building. On the first processed dataset (E3-Nic) molecular replacement tool from Phenix1.20 using the ECD of the Epi-bound nanodisc structure (PDB 7KOQ) was used to place α7. Phenix1.20 chain tracing was used to de novo build the five E3 molecules from their amino-acids sequence, except for the C-terminal Histag, which is flexible and forms a poorly resolved density on the apex of the structure. Model was modified in Coot and refined in Phenix1.20 using secondary structure and Ramachandran restraints. Using the glycosylation tool from Coot, we were able to build the first sugars of the glycosylations on Asn23 and Asn67 and two sugars on Asn110, but densities suggest denser glycosylation trees (Supplementary Fig. 7). The two cysteine disulfide bonds in E3 were well resolved, as was the one in the α7 Cysloop. However, the two cysteine side chains from the canonic disulfide bond of the C-loop (Cys189/Cys190), although well resolved, do not appear to bond in the density. This loop is exposed to the solvent, and the disulfide bond is known to be sensitive to radiation damage (Rhaman et al.[38], Noviello et al.[6], Hattne et al.[39]); we thus omitted the bond in the structure. For the remaining datasets, the resulting E3-Nic structure was used for molecular replacement, the sequence of the nanobodies was further adjusted for C4, and structures were refined using Coot and Phenix and validated using the MolProbity Phenix validation tool.

Structure analysis was performed using Pymol, ChimeraX and PISA from the CCP4 suite.

### Real-time surface plasmon resonance assays

SPR assays were performed on a Biacore T200 instrument (Cytiva) equilibrated in buffer A pH8 at 25 °C.

Surface preparation: Carboxy-methylated dextran CM5 sensorchips (Cytiva) were functionalized by immobilizing covalently the anti-Rho tag monoclonal antibody 1D4 (50 μg/ml at pH4) through amide bonds at a density of 13,000–15,000 resonance units (RU, 1RU ≈ 1 pg/mm2). α7-nanodiscs (10 μg/ml) were then captured on the 1D4 surfaces at a density of 1300-1500RU.

Binding assays: the association and dissociation kinetic properties of nanobodies/α7 complexes were determined by injecting nanobodies in single cycle kinetics mode (five increasing concentration injections of 600 s each at 30 μl/min on both α7/Rho1D4 and reference Rho1D4-only surfaces. This was followed by a final dissociation phase of 1800s. Three-fold dilution series were used for each nanobody (100–1.23 nM and 20–0.25 nM for E3 and its mutants; 6.66–0.08 nM for C4). The specific SPR signals were analyzed using the Biacore T200 evaluation software (Cytiva), yielding association ($k_{on}$) and dissociation ($k_{off}$)rates and equilibrium dissociation constants ($K_d$) for each nanobody/α7 complex.

### Immunofluorescence

HEK293 cells were cultured on poly-D-lysine (Millipore) coated glass coverslips, according to the manufacturer's recommendations. The hα7-nAChR-IRES-eGFP transfer plasmid or its mutated version (α3MIR or S25A) was co-transfected with NACHO, Ric-3 and SAT1 plasmids (1:1:1:1 ratio) using and the JetPrime transfection reagent (Polyplus), according to manufacturer instructions.

Then, 48–72 h after transfection, cells were fixed with 4% PFA and permeabilized for 2 min with a solution of ethanol/methanol (1:1). Nonspecific binding was blocked with 10% BSA in PBS for 5 min at room temperature. The nanobodies-Fc constructs were diluted to 5 μg/ml in 10% BSA in PBS and incubated with the coverslips for 2 h at room temperature.

Proper expression of α7-nAChRs was verified by a mouse anti-Rho1D4. Anti-human IgG and anti-mouse IgG coupled to Alexa647 (ThermoFisher) were diluted in PBS-BSA.

Coverslips were mounted on slides after Prolong-DAPI staining (Invitrogen) and visualized using epi-fluorescence at constant exposure times. All experiments were reproduced ≥4 times.

### Two-electrode voltage-clamp electrophysiology

*Xenopus laevis* oocytes were obtained from EcoCyte Bioscience, Germany, and from Tefor Paris-Saclay UAR2010 and maintained in modified Barth's medium (87.34 mM NaCl, 1 mM KCl, 0.66 mM NaNO₃, 0.75 mM CaCl₂, 0.82 mM MgSO₄, 2.4 mM NaHCO₃, 10 mM HEPES pH 7.6). Defolliculated oocytes were submitted to intranuclear injection of ~2–6 ng of hα7-nAChR-IRES-eGFP transfer plasmid and kept at 18 °C for 2–3 days before recording.

Recordings were performed with a Digidata 1550 A digitizer (Molecular Devices), an Axon Instruments GeneClamp 500 amplifier (Molecular Devices), an automated voltage-controlled perfusion system which controls an 8-port and a 12-port electric rotary valve (Bio-Chem Fluidics) both connected to a 2-way 4-port electric rotary valve (Bio-Chem Fluidics) and the pClamp 10.6 software (Molecular Devices). Some additional recordings were performed as described previously[40].

Oocytes were perfused with Ringer's buffer (100 mM NaCl, 2.5 mM KCl, 10 mM HEPES, 2 mM CaCl₂, 1 mM MgCl₂, pH 7.3). Nanobodies solutions were applied after dilution in Ringer's buffer, and all currents were measured at −60 mV. For potentiation assays, oocytes were perfused with Ringer's buffer for 30 s, then 5 s with 30 μM ACh (or 0,3 μM for L9′T) followed by 2- or 3-min wash and another 5 s ACh application. The nanobody solution was then perfused for 30 s (or 10 s for L9′T) followed by 5 s of 30 μM ACh and 2- or 3-min wash and another 5 s ACh application. For ACh dose-response curves, the nanobody solution was perfused for 15 s before each ACh application.

Recordings were analyzed using ClampFit and GraphPad Prism. Measurements were performed at the peak of the response. For the statistical comparisons using GraphPad Prism, we performed Student's unpaired *t*-test.

### Reporting summary

Further information on research design is available in the Nature Portfolio Reporting Summary linked to this article.

## Data availability

The cryo-EM maps have been deposited in the Electron Microscopy Data Bank (EMDB) under accession codes EMDB-16513 (C4 Apo), EMDB-16534 (C4 Nic), EMDB-16598 (E3 Apo), EMDB-16665 (E3 Nic) and EMDB-16666 (C4 partial). The structural coordinates have been deposited in the RCSB Protein Data Bank (PDB) under the accession codes 8C9X (C4 Apo), 8CAU (C4 Nic), 8CE4 (E3 Apo), 8CI1 (E3 Nic) and 8CI2 (C4 partial). Source data are provided with this paper.

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

## Acknowledgements

This work was supported by an ERC (Grant no. 788974, Dynacotine) to P.J.C. and M.S.P., an Institut Pasteur Axe3 Seed grant to M.S.P., iNEXT-Discovery funding (PID: 22438) to M.S.P., the Foundation de la Recherche Médicale (Equipe FRM DEQ20140329497) to P.J.C., the Agence Nationale de la Recherche (Grant ANR-21-CE37-0026, Nicoptotouch) to P.J.C. and G.A., and the Institut National du cancer INCA to G.D.B. The authors thank the Nanoimaging Core Facility (Institut Pasteur) and Simon Fromm (EMBL-HD) for their precious help in data collection and H. Nury, A. Menny and N. Wolff for discussions and critical reading of the manuscript.

## Author contributions

M.S.P. and P.J.C. conceived and supervised the study. N.B. and S.P. prepared lentiviruses with U.M.'s supervision. P.L. and G.A. produced nanobodies. M.S.P. and N.B. set up and performed protein production, cryoEM sample preparation and data collection. M.G., G.P.A. and F.B. helped in early biochemistry, electron microscopy and data processing, respectively. M.S.P. processed the cryoEM data and solved the structures. N.B. and P.E. performed SPR experiments. G.D.B. performed electrophysiology experiments. All authors helped in the analysis and interpretation of the results. M.S.P. and P.J.C. wrote the manuscript with contributions from all authors.

## Competing interests

G.A., P.J.C., P.L., M.S.P. and N.B. are inventors of patent application US 63/383,099 that covers the nanobodies and therapeutic uses thereof. The remaining authors declare no competing interests.
