## [Peer Review File · Nature Communications]

An original potentiating mechanism revealed by the cryo-EM structures of the human $\alpha 7$ nicotinic receptor in complex with nanobodiesReviewers' Comments:

Reviewer #1:

Remarks to the Author:

The authors previously (in BioRxiv) reported nanobodies specific for the $\alpha 7$ homo-pentameric receptor, with some being silent while other have PAM activity. The present study was aimed at understanding to binding mode, and the way they can potentiate receptor activity. To reach a very detailed view of their mode of action, the authors solved the cryoEM structure of 5 different complexes with bound silent (C4) of PAM (E3) nanobodies, at either saturating on non-saturating conditions. They first characterized the binding mode of the C4 to both the apo and nicotine bound form of the receptor. When using highly saturating concentrations of nanobodies, the cryoEM structures revealed one nanobody per subunit with an axial symmetry, similar to the organization of the five ECDs. The binding modes are almost identical on the apo and nicotinic bound receptors, despite the clear observation that these two forms are resting and active-or-desensitized, then confirming the ability of this nanobody to bind to both forms of the receptor, validating its lack of effect. Such a 5-5 stoichiometry is different from that estimated from the binding data determined by plasmon resonance measurements that indicated a B_{max} of 3 C4 per pentameric receptor. The authors then determined the structure of the pentameric receptor with a lower Nb concentration and observed three nanobodies per pentamer, well organized on the top of subunits A, B and D. Such structure also reveals a different positioning of the nanobody when binding alone or in pair, compared to when saturated, indicating a two-steps binding mode. When analyzing the E3 binding, they found a similar binding epitope as that of C4, with 5 nanobodies per pentamer. Most interestingly, the receptor is found in an active-like form with the orthosteric site entry closed and a potential cation inside the binding site, consistent with this nanobody stabilizing an active conformation of the receptor. Here again, a 5-5 stoichiometry is observed, in contrast to a B_{max} of 3 per pentamer determined by the binding studies. It is then proposed that only the 5-5 stoichiometry stabilizes the active receptor, as a high concentration of E3 is necessary for this effect, more than 1000-fold higher than its K_d . The authors then identified the important role of a glycan interacting with E3 in its potentiating effect. Such data are clear, very convincing, reporting a quite detailed view of the nanobody mode of action on such a pentameric receptor. I only have a few comments that the authors could consider to improve their manuscript.

1) It is quite surprising that E3 which exerts a clear PAM action did not increase agonist affinity. I was wondering whether the authors did test high enough concentration of E3 as only those can yield to a 5-5 stoichiometry that apparently is needed for the PAM effect. Such an increase in affinity would be consistent with the observation of the closure of the binding site entry.

2) According to the official nomenclature of allosteric compounds, silent allosteric compounds are not called "modulators" as, on their own, they have no effect on the orthosteric ligand action. IUPHAR recommends the use of "silent allosteric ligands" (SAL), instead of silent allosteric modulators (See Christopoulos et al., Pharm Reviews 2014)

3) Are the effect of PNU and E3 similar ? are they additive ? do they synergize ?

Reviewer #2:

Remarks to the Author:

The human $\alpha 7$ nAChR plays critical roles in memory, attention and learning. Agonists and allosteric modulators of $\alpha 7$ nAChR have been developed for potential treatment of neurological and psychiatric disorders, such as Alzheimer's disease, schizophrenia, and so on. In this manuscript, the authors reported five cryoEM structures of $\alpha 7$ nAChR in complex with two nanobodies, which named E3 and C4. Their electrophysiological results showed that E3 acted as a type I PAM, which potentiated ACh-gated currents, while C4 acted as a silent allosteric modulator. The authors wish to understand the allosteric mechanism of these nanobodies to $\alpha 7$ nAChR by solving the cryoEM structures of the complexes. Apart from the $\alpha 7$ /C4/nicotine complex which including full-length $\alpha 7$, the other four

complexes have ICD-truncated $\alpha 7$ ($\alpha 7\Delta$ ICD) instead. However, in all of the five complexes structures, the TMD and ICD are not resolved. Recently the $\alpha 7$ nAChR structures in different conformations have been reported in detergent or in nanodiscs by two independent groups. The structures of TMDs and ICDs in both papers are well solved. In this manuscript, the purified $\alpha 7$ nAChR was also reconstituted in nanodiscs. It is unusual that the TMDs are invisible. I have concern about the proper function of the purified proteins.

Due to lack of the TMD structures, the authors define the active or desensitized conformations by comparing the loopC capping and Cysloop motion in these complexes with previous published $\alpha 7$ nAChR cryoEM structures. I suspect if it is reasonable. Generally, we define the different conformations (close, open, or desensitized) of the ion channel by measuring the pore diameters. It is inappropriate to describe the conformation of a channel without any structure information of the transmembrane domain. In its present form, I do not think the key conclusions are justified.

An interesting finding in the manuscript is the conformation of the $\alpha 7\Delta$ ICD/E3 structure, which displays similar agonist-bound conformations, including loopC capping and Cys-loop outward motion. Upon agonist binding, the ligand makes contacts with the loopA, C and E, mainly through the aromatic residues in these loops. The E3 is not bound in the orthosteric site, and only a well-defined small spherical density was found in the cavity. The authors speculate the density to be water or a Na^+ , but without further verification. What triggers the loopC capping in the absent of agonist in the E3/ $\alpha 7\Delta$ ICD complex? I would suggest the authors further investigate the allosteric modulation mechanism of E3/ $\alpha 7$ that the readers are interested to know.

Minor comment,

Line 377, "Interestingly, the unsharpened map of the E3/C4-bound structures suggests that glycosylation trees could substantially mask the orthosteric site (Supplementary Figure 7) ..."
It is better to highlight the glycosylation position in the map.

Text and corresponding figures are mismatch, and some examples are list below.

Line 307, Supplementary Figure 11->12

Line 343, Figure6 - > Supplementary Figure 4g

Typo.

Line 283, Nter -> N-terminal

Line 339, his -> this

Line 339, co-structures?

REVIEWER COMMENTS

Reviewer #1 (Remarks to the Author):

1) It is quite surprising that E3 which exerts a clear PAM action did not increase agonist affinity. I was wondering whether the authors did test high enough concentration of E3 as only those can yield to a 5-5 stoichiometry that apparently is needed for the PAM effect. Such an increase in affinity would be consistent with the observation of the closure of the binding site entry.

The reviewer is right, and we have investigated this further.

First, for information, our article describing the generation and characterization of the C4 and E3 nanobodies is now published (Li et al, Cell Mol Life Sci. 2023 May 25;80(6):164. doi: 10.1007/s00018-023-04779-8).

In the present paper, we provide compelling evidence that E3 at sub-saturating concentrations promotes a PAM effect, and at saturation or near saturation (5-5 stoichiometry) stabilizes an agonist-bound conformation of the ECD in cryo-EM, suggesting a possible agonist activity (Ago-PAM) in electrophysiology. To investigate this possibility, we challenged oocytes up to a concentration of 10 μ M E3 in the absence of ACh, but did not observe direct activation. Of note, performing electrophysiology experiments with such high concentration of E3 (>1 μ M) is very costly in terms of nanobody requiring several mg of protein per condition. However, even if 10 μ M would be sufficient to promote significant activation, the fact that E3 also triggers desensitization, combined with its slow association to the receptor, would probably mainly desensitize the receptor in the recording conditions. This was already stated in the manuscript.

The weak effect of E3 partial binding on the apparent affinity of ACh in electrophysiology (described in Li et al 2023) was also surprising to us. Following the referee comments, we reproduced these experiments under relatively high concentrations of E3 (15'' pre-applications of 1 μ M of E3), again showing no significant change in the apparent affinity for ACh. Interestingly, while potentiation is clearly observed in the first portion of the ACh concentration-response curve (below its EC₅₀), no potentiation is observed above. We suspected that rapid desensitization of α 7 combined with the low temporal resolution of oocyte TEVC could mask the peak potentiation of E3 at high ACh concentrations.

To test this possibility, we used the L247T mutant that displays much slower desensitization kinetics. On this mutant, 15'' 1μM pre-application of E3 does potentiate currents at most ACh concentrations.

Data show a significant increase in ACh apparent affinity in the presence of E3 (17 ±20 μM versus 4 ±1 μM in the presence of E3, n=10, p<0.05 in an unpaired t-test). In conclusion, the oocyte TEVC system on the fast-desensitizing receptor makes it difficult to determine the precise changes in ACh affinity on the wild-type α7, but data obtained on the L247T does show that E3 significantly increases affinity of the receptor for ACh.

These data are now shown in figure 6 and a new Supplementary Figure 14. They are also presented in a new results section, that describes all electrophysiological experiments on the L247T mutant (section 6/, placed after the description of the cryo-EM data).

2) According to the official nomenclature of allosteric compounds, silent allosteric compounds are not called "modulators" as, on their own, they have no effect on the orthosteric ligand action. IUPHAR recommends the use of "silent allosteric ligands" (SAL), instead of silent allosteric modulators (See Christopoulos et al., Pharm Reviews 2014)

We replaced SAM by SAL throughout the whole text.

3) Are the effect of PNU and E3 similar? are they additive? do they synergize?

We performed additional experiments to investigate this aspect of E3 pharmacology. The effect of PNU on α7 is different from E3 since it impairs desensitization while E3 has low/no effect on desensitization in TEVC. Still, we investigated if their peak potentiation is additive or synergetic by comparing potentiation of 30μM ACh of α7FLcryo response after 30sec pre-application of E3 (300nM), moderate concentration of PNU-120596 (0.5μM) or both.

We observed an average fold potentiation of 1.6 for E3, 4.6 for PNU and 6.7 for them both, clearly showing additive effects in these conditions.

These data are now shown in Supplementary Figure 1 and presented in the results section.

Reviewer #2 (Remarks to the Author):

However, in all of the five complexes structures, the TMD and ICD are not resolved. Recently the $\alpha 7$ nAChR structures in different conformations have been reported in detergent or in nanodiscs by two independent groups. The structures of TMDs and ICDs in both papers are well solved. In this manuscript, the purified $\alpha 7$ nAChR was also reconstituted in nanodiscs. It is unusual that the TMDs are invisible. I have concern about the proper function of the purified proteins.

Indeed, two independent groups previously described $\alpha 7$ structures in either saposin/azolectin nanodiscs or DDM micelles, and this was explicitly stated and discussed in the present manuscript. Of note, in saposin/azolectin nanodiscs, the local resolution of all structures is much lower at the TMD/ICD as compared to the ECD (as discussed and presented in the Fig S3 in the Noviello paper). This is most evident for the “open” structure, that shows poor resolution at the lower part of the TMD. In detergent, the local resolution of the TMD is better, but they didn’t catch an open structure of the pore, only a “partially open or desensitized” one, possibly underlying a detergent-specific effect. This clearly shows that in saposin nanodiscs, the TMD is endowed with relatively high structural flexibility as compared to the ECD.

In our MSP/brain lipids nanodiscs system, the TMD is endowed with even higher structural flexibility precluding model building. Thus, the nanodiscs system we used ended up not to be optimal for TMD building, but it is worth noting that our work was initiated before the Noviello and Zhao papers were released.

The referee is thus right in stating that the function of the purified protein is a concern. Analysis of the C4-Apo structure in one hand, and of the C4-Nic and E3-Nic structures in the other hand, shows conformations of the ECD nearly identical to the Bgt/Apo and agonist-bound structures, respectively. To illustrate better this point, we now included in the manuscript a revised version of supplementary figure 8 (shown below) where the whole ECD of ours and previous structures in nanodiscs are superimposed. It shows that C4apo near perfectly aligns with Bgt-nd, but not with Epi-nd and Epi/PNU-nd where the lower half of the beta-sandwich undergoes a global tilt, and where loop-C is capping. Conversely, C4nic aligns near perfectly to the Epi-nd and Epi/PNU-nd but not with Bgt-nd. Altogether, **in our nanodisc system, the ECD undergoes intact agonist-elicited motions. On the other hand, we don’t know if the ECD is properly coupled to the TMD and if its motions are properly transmitted to the ion channel.**

Due to lack of the TMD structures, the authors define the active or desensitized conformations by comparing the loopC capping and Cysloop motion in these complexes with previous published $\alpha 7$ nAChR cryoEM structures. I suspect if it is reasonable. Generally, we define the different conformations (close, open, or desensitized) of the ion channel by measuring the pore diameters. It is inappropriate to describe the conformation of a channel without any structure information of the transmembrane domain. In its present form, I do not think the key conclusions are justified.

In the manuscript, motions of the loopC and the Cys-loop were used as a proxy to give a quantitative evaluation of the ECD conformation, but they do report intact global motions of the ECD as shown above. Still, the referee is right in arguing that we have no indication of the actual state of the receptor in the absence of visualization of the ion channel. We agree that interpretation of our Cryo-EM data in terms of global allosteric state was somewhat overstated.

Throughout the text, we thus more carefully restated the conclusions, refraining to use “close, open and desensitized” conformations that are now replaced by “ECD agonist-bound” and “ECD apo/Bgt” conformations. We also explicitly state in the results and the discussion that the receptor may be decoupled in our MSP nanodisc system. We also moved figure 7, that summarizes the “Hypothetical model for the allosteric mechanism mediating the PAM/agonist activity of E3”, to the supplementary material (Supplementary Figure 15).

Given this more accurate interpretation of the cryo-EM data, the following conclusions are still fully supported by the experimental data:

Concerning cryo-EM data:

- 1/ The C4apo and C4nic co-structures show conformations of the ECD identical to that of previously solved structures, indicating intact arrangement and intact nicotine-elicited reorganization at this level.
- 2/ The structures show that E3 (but not C4) alone stabilizes an agonist-bound conformation of the ECD, demonstrating a direct allosteric action of this PAM in the absence of nicotine.
- 3/ We have identified at high resolution the mode of binding and the common epitope of C4 and E3, involving unique pentamerization of the nanobodies at the top platform of the receptor, away from the TMD. Structures show that different clusters of interactions are involved in E3 versus C4 binding.

Concerning functional (immunofluorescence, SPR and electrophysiological) data:

- 4/ Mutation of the receptor epitope alters nanobody binding.
- 5/ Mutation of residues of E3 specifically (as compared to C4) involved in receptor binding alter its PAM effect.
- 6/ E3 alone does significantly activate a weakly-desensitizing mutant (L247T) of the receptor.

Overall, we believe that combined structural and functional data support the major conclusions of the paper.

An interesting finding in the manuscript is the conformation of the $\alpha 7\Delta$ ICD/E3 structure, which displays similar agonist-bound conformations, including loopC capping and Cys-loop outward motion. Upon agonist binding, the ligand makes contacts with the loopA, C and E, mainly through the aromatic residues in these loops. The E3 is not bound in the orthosteric site, and only a well-defined small spherical density was found in the cavity. The authors speculate the density to be water or a Na^+ , but without further verification.

In the E3-Nic dataset (2.7 Å resolution), the density is clear in the orthosteric site with the 2 rings of the nicotine molecule clearly visible. At the same contour level and with a similar resolution (2.6 Å), the E3-Apo dataset indeed displays a density compatible to either a small monovalent ion or a water molecule:

Several groups are developing tools to accurately predict tightly coordinated ions and water molecules in very high-resolution structures ($<2 \text{ \AA}$, Zhang, K., Pintilie, G.D., Li, S. *et al.* *Cell Res* **30**, 1136–1139 (2020). Pintilie, G. & Chiu, W. (2021). *Acta Cryst. D* **77**, 1142-1152.). But to the best of our knowledge, no method can be used to accurately discriminate between the two possibilities in our datasets. In addition, we feel that identifying this density would not add much to our study.

What triggers the loopC capping in the absent of agonist in the E3/a7 Δ ICD complex? I would suggest the authors further investigate the allosteric modulation mechanism of E3/a7 that the readers are interested to know.

In the discussion, we stated that:

“The PAM activity is strongly impaired by disrupting a key polar interaction with a sugar from the principal subunit nearby the interface with the complementary subunit, and possibly two putative salt bridges with the complementary n+1 E3 and a7 subunit. We speculate that the binding of E3 constrains the relative arrangement of the principal and complementary subunits of a7, favouring a subunit tilt associated with gating (Figure 7).”

In the published structures, as well as in the present work concerning the ECD, the top platform undergoes subtle motions during activation and desensitization, this is why we refrained to propose an allosteric mechanism that would be highly speculative. Still, the ability of E3 binding at the top platform to drive an allosteric reorganization that eventually leads to the closing of loopC is fascinating. To further investigate this question, we superimposed a dimer of C4apo (resting-like ECD) and of E3apo (agonist-bound-like ECD) structures (exemplified in the figure below). Superimposition was performed

at the level of the N-terminal helix of the $\alpha 7(n)$ subunit, to best visualize the quaternary reorganization between $\alpha 7(n)$ and $\alpha 7(n+1)$ subunits.

We speculate that E3 binding would apply constraints on the interface between subunits (notably through specific interaction or “pressure points” highlighted in the study), favoring subunit tilt that is associated with modest reorganization at the top, and stronger reorganization at the bottom of the ECD. We further speculate that this quaternary reorganization would yield a local conformation of the binding site favoring the capping of the loopC. Further studies, possibly by molecular dynamic simulations, would be needed to investigate this possibility.

Minor comment,

Line 377, “Interestingly, the unsharpened map of the E3/C4-bound structures suggests that glycosylation trees could substantially mask the orthosteric site (Supplementary Figure 7) ...”
It is better to highlight the glycosylation position in the map.

We modified the figure that now include the positions of the glycosylation trees.

Text and corresponding figures are mismatch, and some examples are list below.

Line307, Supplementary Figure11->12

Line 343, Figure6 -> Supplementary Figure 4g

We modified the manuscript accordingly.

Typo.

Line 283, Nter → N-terminal

Line 339, his -> this

Line 339, co-structures?

We modified the manuscript accordingly.

Reviewers' Comments:

Reviewer #1:

Remarks to the Author:

The authors did a great job in clarifying all the points I raised in my review. Congratulation for this nice story.

Reviewer #2:

Remarks to the Author:

In the revised manuscript, the authors have added some electrophysiological results to verify the PAM effect of E3 to $\alpha 7$ receptor. However, they did not improve the structures. Indeed, the TMD in the open-state of $\alpha 7$ is relatively flexible as compared to the ECD. However, the TMD is much more rigid in the apo/resting/desensitized state. As presented in the Noviello paper (reference 6), the local resolution of $\alpha 7$ at the TMD/ICD is low (>3.5 Å) in the open-state. However, in the resting and desensitized state, the local resolution of $\alpha 7$ at the TMD is good enough to build accurate atomic model.

In the previously reported structures of $\alpha 7$ (reference 6 and 7), the complexes of $\alpha 7$ with agonists were both in the desensitized state and the TMD were well-resolved. In the present manuscript, E3 potentiates Ach-gated currents and has no effect on the desensitization process of $\alpha 7$. Based on the fast-desensitized property of $\alpha 7$, both of the "E3-Apo" and "E3-Nic" complex were supposed to be in the desensitized state. Strictly speaking, there was at least a prominent class containing particles in the desensitized state. Unfortunately, there are no structural information about the TMD in above two complex. As for the C4, it acts as a silent allosteric ligand of $\alpha 7$ and has no effect on Ach-gated currents. According to the property of $\alpha 7$, it was almost impossible that the complex "C4-Apo" was in the open state. It is somewhat unreasonable that the TMDs of $\alpha 7$ in all the complex containing E3 or C4 were invisible. Have the authors tried to obtain the C4-Apo structure using the full-length $\alpha 7$ instead of the $\alpha 7\Delta$ ICDcryo construct?

Besides, the cryo-EM structure refinement statistics for all the five structures are not quite optimal. For example, the former four structures have very high clashscore (10.19-20.95), and the ramachandran outlier is 1.6% for the last structure. Generally, a valid outlier must be supported by the experimental data (unambiguously resolved in the map, for instance) and be justified by local chemistry (for example, a strained conformation stabilized by hydrogen bonding) (see Afonine, P.V., et al. (2018). *Acta Crystallogr D Struct Biol*, 74(9): 814-840). In the "C4partical-Apo" data, the model resolution (3.4 Å) was even much higher than the map resolution (4.4 Å), with the same FSC cutoff (0.143). The authors should further improve the models during refinement.

As for the electrophysiological results, there are also some issues. The property of $\alpha 7$ L247T mutant is different from the wild-type $\alpha 7$. Some agonists which potentiate Ach-evoked currents of the L247T mutant inhibit ACh-induced responses of the $\alpha 7$ instead (see Hogg, R. C., et al. (2003). *JBC* 278(29): 26908-26914). In the present manuscript, E3 activates the current of $\alpha 7$ L247T mutant. It does not equal to that E3 can activate the $\alpha 7$. There was not enough structural or functional evidence to support the hypothetical model proposed in the present manuscript (Fig. S15).

Reviewer #1 (Remarks to the Author):

The authors did a great job in clarifying all the points I raised in my review. Congratulation for this nice story.

We thank the reviewer for his positive evaluation.

Reviewer #2 (Remarks to the Author):

In the revised manuscript, the authors have added some electrophysiological results to verify the PAM effect of E3 to $\alpha 7$ receptor. However, they did not improve the structures. Indeed, the TMD in the open-state of $\alpha 7$ is relatively flexible as compared to the ECD. However, the TMD is much more rigid in the apo/resting/desensitized state. As presented in the Noviello paper (reference 6), the local resolution of $\alpha 7$ at the TMD/ICD is low (>3.5 Å) in the open-state. However, in the resting and desensitized state, the local resolution of $\alpha 7$ at the TMD is good enough to build accurate atomic model.

In the previously reported structures of $\alpha 7$ (reference 6 and 7), the complexes of $\alpha 7$ with agonists were both in the desensitized state and the TMD were well-resolved. In the present manuscript, E3 potentiates Ach-gated currents and has no effect on the desensitization process of $\alpha 7$. Based on the fast-desensitized property of $\alpha 7$, both of the “E3-Apo” and “E3-Nic” complex were supposed to be in the desensitized state. Strictly speaking, there was at least a prominent class containing particles in the desensitized state. Unfortunately, there are no structural information about the TMD in above two complex. As for the C4, it acts as a silent allosteric ligand of $\alpha 7$ and has no effect on Ach-gated currents. According to the property of $\alpha 7$, it was almost impossible that the complex “C4-Apo” was in the open state.

It is somewhat unreasonable that the TMDs of $\alpha 7$ in all the complex containing E3 or C4 were invisible.

Have the authors tried to obtain the C4-Apo structure using the full-length $\alpha 7$ instead of the $\alpha 7\Delta$ ICDcryo construct?

The referee correctly points out that, in the two previously published articles, the closed-channel conformations (resting and desensitized) were solved at better resolution than the open-channel state, especially at the TMD when the receptor was reconstituted in saposin/azolectine nanodiscs. If we understand correctly, the referee is suggesting stabilizing these closed conformations to increase the chances to visualize the TMD in our MSP/brain lipids nanodiscs. We essentially had the same reasoning, and following our observation that the TMD was not resolved in C4-Apo with $\alpha 7\Delta$ ICDcryo construct, we stabilized the conformation to a desensitized state by adding nicotine, in combination with $\alpha 7$ FLcryo construct where the TMD is expected to be structurally constrained by the intracellular domain. Unfortunately, this did not improve the resolution of the TMD.

Another way would be to stabilize the structure in a resting-state (Apo) in combination with $\alpha 7$ FLcryo construct. This is an interesting idea that we did not attempt. However, in the Noviello paper they stabilize the resting state with alpha-bungarotoxin, and there is no guaranty that Apo conditions in the absence of alpha-bungarotoxin would achieve a better TMD resolution.

Besides, the cryo-EM structure refinement statistics for all the five structures are not quite optimal. For example, the former four structures have very high clashscore (10.19-20.95), and the ramachandran outlier is 1.6% for the last structure. Generally, a valid outlier must be supported by the experimental data (unambiguously resolved in the map, for instance) and be justified by local chemistry (for example, a strained conformation stabilized by hydrogen bonding) (see Afonine, P.V., et al. (2018). *Acta Crystallogr D Struct Biol*, 74(9): 814-840).

According to the referee suggestions, we refined the models and improved the overall statistics that are displayed in the new Supplementary Table 1.

More specifically, new *clashscore* values are:

- C4-Apo 5.12
- C4-Nic 6.08
- E3-Apo 5.41
- E3-Nic 4.33
- C4partial-Apo 6.38

For C4partial-Apo, we fixed the Ramachandran outliers.

We highlighted those additional changes in green in the manuscript.

In the “C4partial-Apo” data, the model resolution (3.4 Å) was even much higher than the map resolution (4.4 Å), with the same FSC cutoff (0.143). The authors should further improve the models during refinement.

Indeed, the original map resolution for this dataset is 4.4Å. To assist in model building, we used the density modification tool from Phenix described here:

Improvement of cryo-EM maps by density modification. T.C. Terwilliger, S.J. Ludtke, R.J. Read, P.D. Adams, and Afonine. Nature Methods 17, 923-927 (2020).

This tool allows improving the density map considering the protein sequence or an initial model. In our case the tool improved by ~0.4Å the map resolution :

In some areas the density became much easier to interpret at equivalent contour values:

The resulting model after refinement has a 3.4 Å resolution as calculated by Phenix. We agree that this could lead to confusion, and, to more accurately reflect the quality of the experimental data, we chose to keep the resolution value of the original map since it is unbiased by the Phenix tool. We added some precision in the text and Supplementary Table 1 to clarify this to the reader. Of note, this C4partial-Apo structure was interpreted only in terms of the global orientation of the bound nanobodies, relying on the tilt of the beta-sandwich core that is already well resolved in the original map. The atomic details of this structure were thus not used to support the main conclusions of the paper.

We highlighted those additional changes in green in the manuscript.

As for the electrophysiological results, there are also some issues. The property of a7 L247T mutant is different from the wild-type a7. Some agonists which potentiate ACh-evoked currents of the L247T mutant inhibit ACh-induced responses of the a7 instead (see Hogg, R. C., et al. (2003). *JBC* 278(29): 26908-26914). In the present manuscript, E3 activates the current of a7 L247T mutant. It does not equal to that E3 can activate the a7. There was not enough structural or functional evidence to support the hypothetical model proposed in the present manuscript (Fig. S15).

The referee is right in stating that observations on the strong gain-of-function a7L247T mutant may not be applicable to the wild-type receptor. The hypothetical model included as supplementary figure S15 aimed to propose one possible interpretation of the whole set of data. However, we agree that other models might be equally conceived. We thus removed Fig. S15 as suggested. In addition, we now explicitly state in the whole manuscript that the partial agonist action of E3 was observed specifically on a7L247T mutant.

We highlighted those additional changes in green in the manuscript.

Reviewers' Comments:

Reviewer #2:

Remarks to the Author:

There are some minor issues about this manuscript.

1. In Supplementary figure 4a-e, the fitting curves were not quite fit with the SPR data. I doubt if the function used in the manuscript is the optimum one. Besides, the authors should also show the values of R squared in the table.
2. In Figure 4d, there is a redundant "d" in the bottom figure.
3. In Figure 6a-d, it would be clearer to have the a7 L247T mutant labeled beside the figures, to distinguish with the a7 WT.

Reviewer #2 (Remarks to the Author):

There are some minor issues about this manuscript.

1. In Supplementary figure 4a-e, the fitting curves were not quite fit with the SPR data. I doubt if the function used in the manuscript is the optimum one. Besides, the authors should also show the values of R squared in the table.

The referee correctly points out that some fits of the SPR data are not optimal. We now include the Chi^2 of the fits in the Table of the Supplementary Figure 4. All appear reasonable (Chi^2 being less than 10% of the Rmax) except for E3 WT of prep 2 that shows a Chi^2 slightly above this threshold. The fitting could possibly be improved by using more complex equations implementing for instance several k_{on} and/or k_{off} values as suggested by the possible negative cooperativity of nanobody binding discussed at the end of the main text. However, such fitting would require performing a systematic set of SPR recordings at various nanobodies concentrations that is beyond the scope of this work. Given these limitations, SPR data indicate nanomolar affinity of both nanobodies and weak effect of E3 mutants E110Q and R56A on binding.

	$\alpha 7$ prep 1		$\alpha 7$ prep 2			
	E3 WT	E3 WT + 100 μ M Nicotine	E3 WT	E3 E110Q	E3 R56A	C4
k_{on} ($10^6 \text{ M}^{-1} \cdot \text{s}^{-1}$)	1.6 \pm 0.6	1.2 \pm 0.5	11 \pm 0.02	18.2 \pm 0.06	7.0 \pm 0.15	1.2 \pm 0.05
k_{off} (10^{-2} s^{-1})	1.3 \pm 0.5	0.85 \pm 0.3	1.9 \pm 0.01	1.1 \pm 0.01	1.0 \pm 0.02	0.024 \pm 0.002
Kd (nM)	8.0 \pm 3	7.4 \pm 0.3	1.0 \pm 0.01	0.62 \pm 0.02	1.5 \pm 0.01	0.20 \pm 0.26
Rmax	137 \pm 50	151 \pm 22	90 \pm 2	65 \pm 1	86 \pm 1	78 \pm 14
Chi^2	5.9 \pm 0.5	12 \pm 1.9	23.2 \pm 0.8	5.3 \pm 0.01	5.4 \pm 0.1	5.1 \pm 2.7

2. In Figure 4d, there is a redundant “d” in the bottom figure.

We removed the redundant d.

3. In Figure 6a-d, it would be clearer to have the $\alpha 7$ L247T mutant labeled beside the figures, to distinguish with the $\alpha 7$ WT.

We labeled the figure accordingly.